# Rejuvenating conventional dendritic cells and T follicular helper cell formation after vaccination

**Marisa Stebegg[1†], Alexandre Bignon[1†], Danika Lea Hill[1], Alyssa Silva-Cayetano[1], Christel Krueger[2], Ine Vanderleyden[1], Silvia Innocentin[1], Louis Boon[3], Jiong Wang[4], Martin S Zand[4], James Dooley[5], Jonathan Clark[6], Adrian Liston[1], Edward Carr[1,7], Michelle A Linterman[1]\***

[1]Laboratory of Lymphocyte Signalling and Development, Babraham Institute, Cambridge, United Kingdom; [2]Epigenetics Programme, Babraham Institute, Cambridge, United Kingdom; [3]Bioceros BV, Utrecht, Netherlands; [4]Division of Nephrology, Department of Medicine and Clinical and Translational Science Institute, University of Rochester Medical Center, Rochester, United States; [5]Autoimmune Genetics Laboratory, VIB and University of Leuven, Leuven, Belgium; [6]Biological Chemistry, Babraham Institute, Cambridge, United Kingdom; [7]Department of Medicine, University of Cambridge, Cambridge, United Kingdom

**Abstract** Germinal centres (GCs) are T follicular helper cell (Tfh)-dependent structures that form in response to vaccination, producing long-lived antibody secreting plasma cells and memory B cells that protect against subsequent infection. With advancing age the GC and Tfh cell response declines, resulting in impaired humoral immunity. We sought to discover what underpins the poor Tfh cell response in ageing and whether it is possible to correct it. Here, we demonstrate that older people and aged mice have impaired Tfh cell differentiation upon vaccination. This deficit is preceded by poor activation of conventional dendritic cells type 2 (cDC2) due to reduced type 1 interferon signalling. Importantly, the Tfh and cDC2 cell response can be boosted in aged mice by treatment with a TLR7 agonist. This demonstrates that age-associated defects in the cDC2 and Tfh cell response are not irreversible and can be enhanced to improve vaccine responses in older individuals.

**\*For correspondence:**
Michelle.Linterman@babraham.ac.uk

[†]These authors contributed equally to this work

**Competing interests:** The authors declare that no competing interests exist.

## Introduction

Successful T cell-dependent vaccines induce the formation of germinal centres (GCs) in secondary lymphoid organs. The GC is a specialised microenvironment that produces long-lived humoral immunity that can provide protection against subsequent infection (*Vinuesa et al., 2016*). Despite the success of T cell-dependent vaccines to date in children and younger adults, vaccination is less effective in older persons (*Govaert et al., 1994*). This has been proposed to be a result of a deterioration in the magnitude and quality of the GC response (*Aberle et al., 2013*; *Linterman, 2014*; *Gustafson et al., 2018*). Within the GC, antigen-specific GC B cells clonally expand and somatically hypermutate the genes encoding their B cell receptor. This mutational process, coupled with subsequent affinity-based selection, results in the emergence of plasma cells and memory B cells that bind antigen with improved affinity (*Stebegg et al., 2018*). The GC response is a highly collaborative process that requires multiple cell types to interact at the right place and the right time: therefore, defects in one or more of these cell types could underlie the poor GC response observed after vaccination of older individuals. Both T cell-intrinsic defects and changes in the microenvironment have

been implicated in the diminished GC response observed in older individuals (*Yang et al., 1996*; *Garcia and Miller, 2001*; *Eaton et al., 2004*; *Lefebvre et al., 2012*; *Linterman, 2014*; *Sage et al., 2015*; *Gustafson et al., 2018*; *Nikolich-Žugich, 2018*), but the precise cellular and molecular changes that cause the age-dependent defects in the GC response remain unclear.

The GC response is absolutely dependent on T cell help. This is delivered by a specialised subset of CD4$^+$ T cells called T follicular helper (Tfh) cells, which provide survival and differentiation signals to GC B cells (*Vinuesa et al., 2016*). Tfh cell differentiation from naïve T cells is initiated by priming by dendritic cells (DCs), which provide three signals to support Tfh cell formation (*Krishnaswamy et al., 2018*): peptide-MHC-II, co-stimulation in the form of the CD28 ligands CD80 and CD86, and cytokines such as IL-6, IL-12 and IL-27 (*Eddahri et al., 2009*; *Vinuesa et al., 2016*; *Webb and Linterman, 2017*). Several DC subtypes can initiate Tfh cell differentiation (*Chakarov and Fazilleau, 2014*; *Dahlgren et al., 2015*; *Yao et al., 2015*; *Levin et al., 2017*; *Barbet et al., 2018*), but migratory conventional type 2 dendritic cells (cDC2s) have been proposed as the dominant Tfh cell-priming DC subset after vaccination (*Krishnaswamy et al., 2017*; *Krishnaswamy et al., 2018*; *Durand et al., 2019*). Tfh precursor cells form after interactions with cDC2s, and subsequent interactions with B cells support completion of Tfh cell differentiation and localisation to the GC (*Goenka et al., 2011*; *Baumjohann et al., 2013*; *Barnett et al., 2014*). Once within the GC, Tfh cells act as the gatekeepers of the B cell response by providing survival signals to the highest-affinity GC B cells, thereby regulating which B cells are able to exit the GC as long-lived plasma cells or memory B cells (*Victora et al., 2010*; *Gitlin et al., 2014*; *Goenka et al., 2014*; *Vinuesa et al., 2016*). The selective help that Tfh cells provide is essential for the quality of the GC response (*Victora et al., 2010*), and for preventing the emergence of autoreactive B cell clones from the GC as long-lived plasma cells or memory B cells (*Linterman et al., 2009*). An impairment in the formation of Tfh cells has been implicated in the defective GC response in aged mice (*Lefebvre et al., 2012*; *Sage et al., 2015*). In this study we sought to identify the mechanism that underlies poor Tfh cell formation in ageing, and to test whether it is possible to reverse the age-dependent defects in Tfh cells.

Here, we report that the circulating counterparts of GC-Tfh cells are diminished in older persons after seasonal influenza vaccination. This impairment in Tfh cell differentiation in humans could be recapitulated in 2-year-old mice, which had fewer antigen-specific Tfh cells in the draining lymph node (LN) after immunisation than younger adult mice. The defective Tfh cell response in aged mice was linked with impaired T cell priming by cDC2s: after immunisation, fewer antigen-bearing cDC2s were found in the draining LN of aged mice, and those that were present had reduced expression of the co-stimulatory ligands CD80 and CD86. Transcriptional profiling of cDC2s from aged mice revealed that they had a defective response to type I interferon (IFN-I) due to the reduced induction of *Ifnb1* after immunisation. Topical application of the TLR7 agonist imiquimod increased the number of antigen-bearing cDC2s, their expression of CD80 and CD86, and restored the formation of antigen-specific Tfh cells in aged animals. This demonstrates that age-associated defects in Tfh cell differentiation are not irreversible and that DCs are a rational target to boost responses to vaccination in older individuals.

## Results

### Circulating Tfh cells are reduced in older persons after vaccination

Circulating Tfh-like (cTfh) cells can be used in humans as a biomarker of concomitant GC reactions after vaccination (*Bentebibel et al., 2013*). In our cohort of 18–36 year-old and 65–75 year-old individuals vaccinated with the 2014–2015 trivalent seasonal influenza vaccine (*Figure 1A*), there was a five-fold reduction in the production of vaccine-specific antibodies in the serum of older persons compared to younger participants (*Figure 1B*), consistent with previous reports (*Govaert et al., 1994*; *Goodwin et al., 2006*; *Sasaki et al., 2011*; *Nakaya et al., 2015*). This diminished antibody production with age was associated with reduced formation of cTfh cells that are transcriptionally and clonally related to *bona fide* GC Tfh cells found in secondary lymphoid organs (*Heit et al., 2017*; *Brenna et al., 2020*; *Hill et al., 2019*). Prior to vaccination, the frequency of CXCR5$^+$PD-1$^{+++}$ CD4$^+$CD45RA$^-$ cTfh cells was similar in the two age groups (*Figure 1C,D*). Seven days after vaccination there were significantly fewer circulating Tfh-like cells in older persons compared to the younger

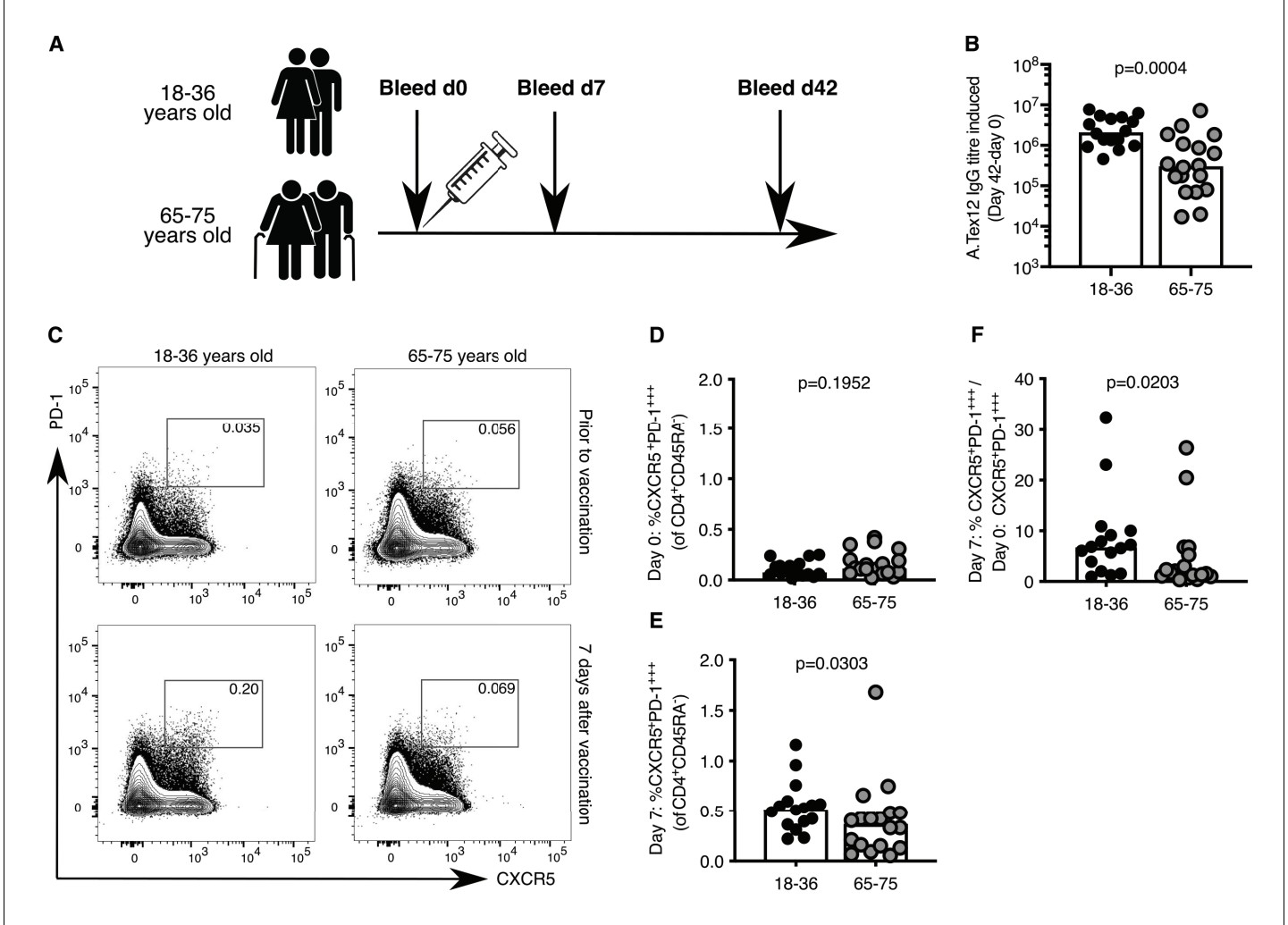

**Figure 1.** Poor induction of circulating Tfh (cTfh)-like cells in older persons upon influenza vaccination. (A) Vaccination and venepuncture schedule for the seasonal influenza vaccination study, 18-36yo n = 16, 65-75yo n = 18. (B) Change in antibody titre of anti-A.Tex12 IgG, an influenza A haemagglutinin (HA), from baseline 42 days after vaccination. Flow cytometric contour plots (C) and quantification of the frequency of CXCR5+PD-1+++ cells amongst CD45RA−CD4+CD3+ cells in the peripheral blood of healthy UK donors at days zero (D) and seven (E) relative to seasonal influenza vaccination. (F) Fold change of cTfh-like cells in the peripheral blood seven days after vaccination over day zero (% CXCR5+PD-1+++CD45RA−CD4+CD3+ on day 7 divided by % CXCR5+PD-1+++CD45RA−CD4+CD3+ on day 0). Bar height corresponds to the median, and each circle represents one biological replicate. P-values generated with a Mann-Whitney test. Data are from one seasonal influenza vaccination cohort from the northern hemisphere in winter 2014–2015.

The online version of this article includes the following source data for figure 1:

**Source data 1.** Poor induction of circulating Tfh (cTfh)-like cells in older persons upon influenza vaccination.

individuals (*Figure 1C,E*), representing an impaired induction after vaccination when normalised to an individual's day 0 baseline (*Figure 1F*). Together, these data indicate that the GC-Tfh cell response to vaccination is impaired in older persons.

## Tfh cell and GC responses are impaired in ageing

A major limitation of human vaccination studies is the difficulty of sampling secondary lymphoid organs after vaccination in large cohorts of people, and thus animal models are invaluable to understand the underlying biology. After vaccination there is an increase in the number of Ki67+Bcl6+-B220+ GC B cells in the draining inguinal LN in both younger 2–3 month-old adult and aged 22–24 month-old mice (*Figure 2A–C*, gating strategy in *Figure 2—figure supplement 1*). However, the number of GC B cells was ten-fold lower in the aged mice ten days after immunisation, compared to

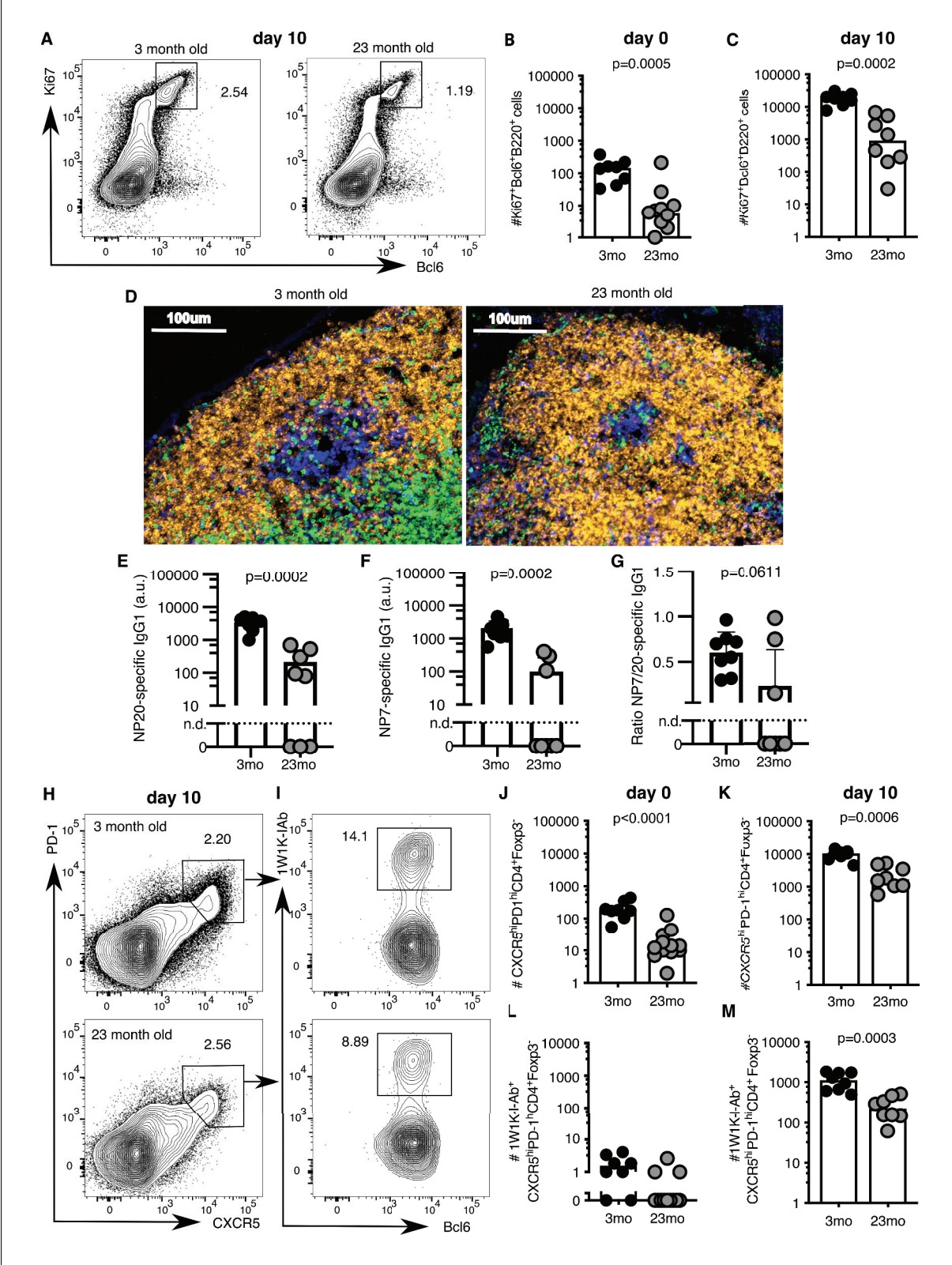

**Figure 2.** Germinal centre (GC) responses are impaired in aged mice. 2–3 month-old adult and 22–24 month-old aged C57BL/6 mice were subcutaneously immunised with NP-1W1K (A-C, E-M) or NP-KLH (D) in Alum. Their draining inguinal lymph nodes (LNs) were analysed by flow cytometry without immunisation (B, J, L), ten days after immunisation (A, C, E-G, H-I, K, M), or after 14 days by confocal imaging (D). (A-C) Representative flow cytometric plots (A) and quantitation (B-C) of B220+Ki67+Bcl6+ GC B cells. (D) Confocal images of draining LNs from 2 to 3 month-

*Figure 2 continued on next page*

*Figure 2 continued*

old and 22–24 month-old mice, taken 14 days after subcutaneous immunisation with NP-KLH. 10 μm LN sections were stained with anti-IgD (orange), anti-CD3 (green), anti-Ki67 (blue) and anti-Foxp3 (pink) antibodies (n = 3–6). (E-G) Levels of NP-specific IgG1 antibodies in the serum of 2–3 month-old and 22–24 month-old mice 10 days after immunisation with NP-1W1K in Alum as determined by ELISA. (E) Serum levels of NP20-specific IgG1 antibodies. (F) Serum levels of high-affinity NP7-specific IgG1 antibodies. (G) Ratio of NP20/NP7-specific IgG1 antibodies in the serum as a measure of antibody affinity maturation. (H-M) Representative flow cytometric plots (H-I) and quantitation (J-M) of CXCR5$^{hi}$PD-1$^{hi}$Foxp3$^-$CD4$^+$ T follicular helper (Tfh) cells (H, J-K) and antigen-specific 1W1K-I-Ab$^+$ Tfh cells (I, L-M). Bar graphs show the results of one of two independent experiments (n = 8–12 per group/experiment). Bar height corresponds to the median, and each circle represents one biological replicate. *P*-values were determined using Mann-Whitney testing. The gating strategy is shown in *Figure 2—figure supplement 1*.

The online version of this article includes the following source data and figure supplement(s) for figure 2:

**Source data 1.** Germinal centre (GC) responses are impaired in aged mice.
**Figure supplement 1.** Gating strategy for GC B and Tfh cell populations.

younger adult mice (*Figure 2C*). This corresponded to a reduction in GC size (*Figure 2D*) and reduced levels of antigen-specific antibodies in the serum of aged mice (*Figure 2E–G*), consistent with previous reports that GC and antibody responses are reduced in magnitude in aged mice (*Kosco et al., 1989*; *Szakal et al., 1990*; *Yang et al., 1996*; *van Dijk-Härd et al., 1997*; *Eaton et al., 2004*; *Linterman, 2014*). This deficiency in the GC response was coupled with reduced numbers of total CXCR5$^{hi}$PD-1$^{hi}$Foxp3$^-$CD4$^+$ Tfh cells prior to, and ten days after immunisation, as well as significantly fewer antigen-specific Tfh cells, as assessed using 1W1K-loaded MHC-II tetramers (*Figure 2H–M*; gating strategy in *Figure 2—figure supplement 1*; key resources are listed in *Supplementary file 1*). This demonstrates that aged mice have impaired Tfh cell formation after immunisation, which recapitulates the age-associated defect in Tfh cell formation observed in humans (*Figure 1*).

## T cell priming is impaired in aged mice

The age-associated deficit in Tfh cells upon immunisation could be due to T cell-intrinsic changes with age, or due to the age of the microenvironment in which the T cells reside. After adoptive transfer of either TCR-transgenic TCR7 or OTII CD4 T cells from 2 to 3 month-old mice into young adult hosts, more than 80% of all transgenic CD4$^+$ T cells had undergone one or more cell divisions after immunisation (*Figure 3A–D*). In contrast, when cells from the same pool of TCR-transgenic T cells were transferred into 22–24 month-old recipient mice, significantly fewer T cells had completed more than one cell division (*Figure 3A–D*). In addition to defects in T cell priming, Tfh cell differentiation of OTII CD4$^+$ T cells isolated from 2 to 3 month-old mice was reduced three-fold ten days after immunisation in aged mice compared to younger recipients (*Figure 3E–F*), which was associated with reduced levels of antigen-specific antibodies in the serum of these mice (*Figure 3—figure supplement 1A–C*). These observations indicate that the aged microenvironment causes impaired early T cell activation and reduced Tfh cell development. These data prompt the hypothesis that the reduced number of Tfh cells induced by immunisation in aged mice may be caused by defective T cell priming.

Several DC subtypes, including LN-resident or migratory type 1 conventional DCs (cDC1s), cDC2s and Langerhans cells, have been implicated in T cell priming (*Chakarov and Fazilleau, 2014*; *Woodruff et al., 2014*; *Yao et al., 2015*; *Levin et al., 2017*; *Barbet et al., 2018*; *Krishnaswamy et al., 2018*). Of these, the migratory cDC2 subset has been suggested as the dominant Tfh cell-priming DC subset early after immunisation (*Krishnaswamy et al., 2017*; *Krishnaswamy et al., 2018*; *Durand et al., 2019*). Consistent with these reports, after subcutaneous immunisation with Eα-GFP the majority of GFP$^+$ antigen-bearing cells in the draining LN were CD11b$^+$ cDC2s (*Figure 3—figure supplement 1D–E*; gating strategy from *Guilliams et al. (2016)*. These cells likely belong to the migratory cDC2s subset, as pertussis toxin treatment, which disrupts G-protein coupled receptor-dependent cell migration, resulted in a reduced number of these antigen-presenting cells in the draining LN (*Figure 3—figure supplement 1F*). One day after immunisation, aged mice had half the number of total and GFP$^+$ CD11b$^+$ cDC2s compared to younger controls (*Figure 3G–H*). This was coupled with the reduced presentation of Eα peptide-MHC-II, despite normal total MHC-II expression on the cell surface (*Figure 3I–J*, *Figure 3—figure supplement 1G*). The expression of the costimulatory ligands CD86, CD80 and the receptor CD40 on the

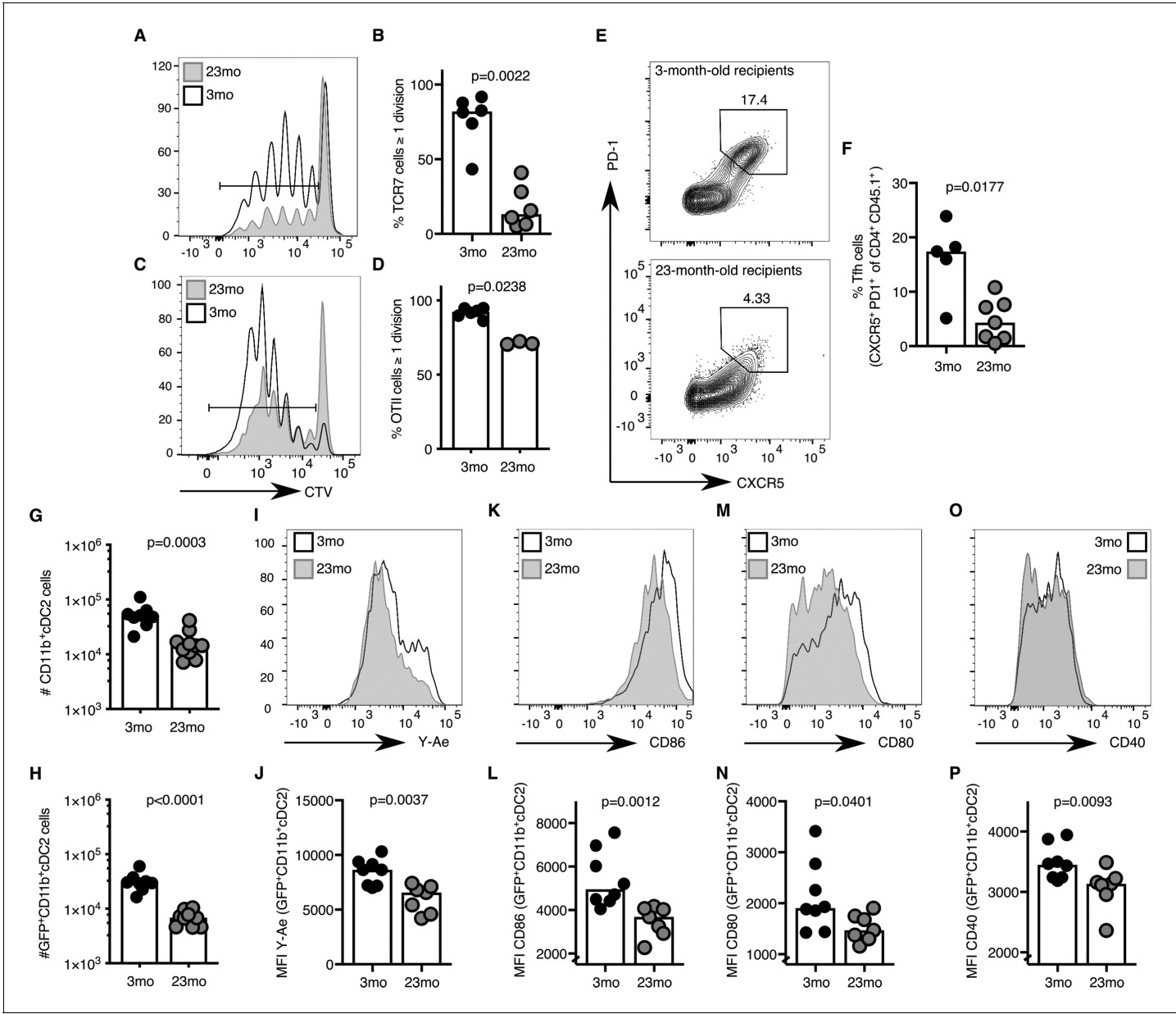

**Figure 3.** Activation of cDC2s and T cell priming are impaired in aged mice. (**A–B**) 1–5 × 10⁶ CellTrace-labelled hen egg lysozyme (HEL)-specific TCR7 (CTV⁺TCRVβ3⁺CD45.1⁺CD4⁺) cells or (**C-D**) 5 × 10⁵ ovalbumin (OVA)-specific (CTV⁺TCRVα2⁺CD45.1⁺CD4⁺) OTII cells were adoptively transferred into 2–3 month-old and 22–24 month-old C57BL/6 recipients, which were subsequently immunised subcutaneously with HEL or OVA in Alum. Cell proliferation of transferred cells was assessed in draining lymph nodes (LNs) by flow cytometry as defined by the serial dilution of CellTrace (CTV) three days after immunisation (n = 3–6 per group/experiment). (**A, C**) Representative flow cytometric plot of CTV levels in CD45.1⁺CD4⁺ cells in 2–3 month-old (white) and 22–24 month-old (grey) C57BL/6 recipients. The gate indicates cells that have undergone one or more divisions. (**B, D**) Percentage of divided CTV⁺CD45.1⁺CD4⁺ cells in 2–3 month-old and 22–24 month-old C57BL/6 recipients. (**E-F**) 5 × 10⁴ OVA-specific (CD45.1⁺TCRVα2⁺CD4⁺) OTII cells were adoptively transferred into 2–3 month-old and 22–24 month-old C57BL/6 recipients, which were subsequently immunised subcutaneously with NP-OVA in Alum in the hind flank to assess T follicular helper (Tfh) cell formation ten days after immunisation (n = 5–6 per group/experiment). Representative flow cytometric plot (**E**) and quantitation (**F**) of CXCR5^hiPD-1^hi Tfh cells formed from CD45.1⁺CD4⁺ OTII cells in 2–3 month-old and 22–24 month-old C57BL/6 recipients. (**G-P**) 2–3 month-old mice and 22–24 month-old were immunised subcutaneously with Eα-GFP in IFA. Antigen-bearing GFP⁺ and antigen-presenting Y-Ae⁺dendritic cells (DCs) in draining LNs were analysed 22 hr after immunisation (n = 7–10 per group/experiment). (**G-H**) Quantitation of total (**G**) and GFP+ (**H**) CD11b⁺ type 2 conventional DCs (cDC2s). (**I-P**) Representative histograms (**I, K, M, O**) and quantitation of median fluorescence intensity (MFI) levels (**J, L, N, P**) of Y-Ae (**I-J**), CD86 (**K-L**), CD80 (**M-N**) and CD40 (**O-P**) on the surface of GFP⁺ CD11b⁺ cDC2s from 2 to 3 month-old and 22–24 month-old mice. Bar graphs show the results of one of at least two independent experiments. Bar height corresponds to the median, and each circle represents one biological replicate. *P*-values were determined using the Mann-Whitney test. Supporting data is shown in *Figure 3—figure supplement 1*.

*Figure 3 continued on next page*

*Figure 3 continued*

The online version of this article includes the following source data and figure supplement(s) for figure 3:

**Source data 1.** Activation of cDC2s and T cell priming are impaired in aged mice.
**Figure supplement 1.** Migratory cDC2s are the main antigen-bearing cells in draining lymph nodes (LNs).
**Figure supplement 1—source data 1.** Migratory cDC2s are the main antigen-bearing cells in draining lymph nodes.

surface of GFP$^+$ CD11b$^+$ cDC2s was also diminished, indicating impaired activation of cDC2s with age (*Figure 3K–P*). Taken together, these data suggest that defective T cell priming in draining LNs of aged mice may be due to impaired antigen presentation and/or co-stimulation from the cDC2 subset.

## The cDC2 response to IFN-I is reduced in aged mice

To determine whether impaired T cell priming in aged mice could be due to reduced antigen presentation by cDC2, we used mice that are haploinsufficient for MHC-II (*H2$^{+/-}$*), to mimic this age-associated phenotype in younger mice. cDC2s from *H2$^{+/-}$* mice present less peptide:MHC-II on their surface 22 hr after immunisation (*Figure 3—figure supplement 1H*), but do not have an impaired capacity to prime CD4$^+$ T cells three days after immunisation (*Figure 3—figure supplement 1I–J*). This indicates that a reduction in antigen-presentation alone does not phenocopy the defect in T cell priming seen in aged mice, suggesting other age-dependent changes drive impaired T cell activation. CD80/CD86 co-stimulation is critical for both T cell priming and Tfh cell differentiation (*Platt et al., 2010*; *Wang et al., 2015b*). By partially blocking these co-stimulatory ligands using a CTLA4-Ig fusion protein in vivo, we observed a dose-dependent decrease in early T cell proliferation (*Figure 3—figure supplement 1K–L*). Moreover, Wang et al. had previously demonstrated that reducing the magnitude of CD28 signalling in vivo impairs Tfh cell differentiation (*Wang et al., 2015a*). This implicates the age-associated reduction in the expression of CD80/CD86 on cDC2s as a likely factor that contributes to impaired T cell priming and Tfh cell formation in aged mice.

To understand the molecular mechanism that underpins the age-associated cellular changes in cDC2s, RNA sequencing was performed on sorted antigen-bearing cDC2s isolated from 2 to 3 month-old and 22–24 month-old mice 22 hr after immunisation with Eα-GFP (gating strategy shown in *Figure 3—figure supplement 1D*). Principal component analysis (PCA) demonstrated distinct clustering of samples with 47% of the variation in this dataset explained by age (*Figure 4A*). Gene set enrichment analysis identified one pathway that was underrepresented in cDC2s from aged mice: the cellular response to IFN-I (*Figure 4B*). This was driven by the reduced expression of the majority of IFN-I inducible genes in aged mice compared to younger adults (*Figure 4C*). A reduction in the IFN-I response in cDC2s was confirmed by quantitative Real-Time PCR (RT-qPCR) in an independent cohort: expression of *Ifit1* and *Mx1*, canonical IFN-I stimulated genes (ISGs), were reduced in aged mice (*Figure 4D–E*). In unimmunised mice, the expression of *Ifnb1* mRNA in the inguinal LN was not detectable. Six hours after immunisation with Eα-GFP *Ifnb1* is expressed in draining LNs, but the induction of *Ifnb1* expression was three-fold lower in 22–24 month-old aged mice compared to 2–3 month-old adult mice (*Figure 4F*).

IFN-I responses are induced by IFNα and IFNβ cytokines which bind the interferon alpha receptor (IFNαR), a heterodimeric complex of IFNAR1 and IFNAR2 chains. Upon ligation of the IFNαR, the Signal Transducer and Activator of Transcription 1 (STAT1) and STAT2 transcription factors are phosphorylated and, in association with the interferon regulatory factor 9 (IRF9), move into the nucleus to promote the expression of ISGs (*Hoffmann et al., 2015*). STAT1 phosphorylation levels and the upregulation of the ISGs *Ifit1* and *Mx1* in cDC2s were intact in aged mice upon ex vivo treatment with low levels of IFNα (*Figure 4G–J*). This indicates that aged cDC2s are capable of responding to IFN-I and do not exhibit cell-intrinsic defects in IFN signalling. Consistent with this, cDC2s derived from bone marrow (BM) stem cells from an aged, 23-month-old mouse transferred into an adult recipient mouse, do not exhibit age-associated defects in number and activation after immunisation when they develop and reside in the microenvironment of the younger host (*Figure 4K–M*). Together, these data suggest that the poor activation of cDC2s in aged animals is not linked with cell-intrinsic defects in their IFN-I response, but rather with impaired early induction of IFN-I in their microenvironment.

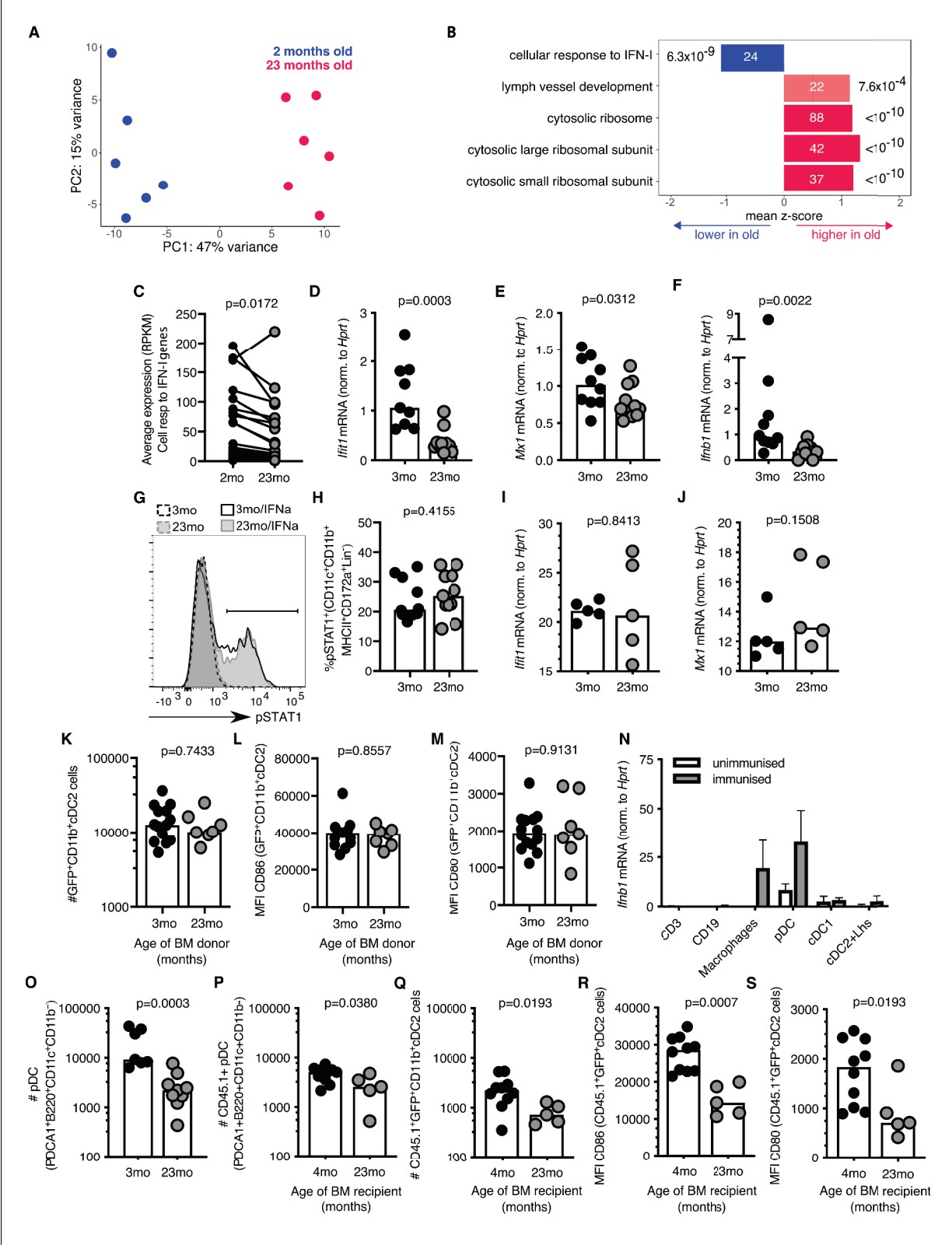

**Figure 4.** Reduced type I interferon (IFN-I) signalling in cDC2s from aged mice. (**A**) Principal component analysis (PCA) of the 1000 genes with the largest variance in sorted GFP⁺CD11b⁺ cDC2s cells from adult 2-month-old (blue) and aged 23-month-old (pink) mice (n = 6 per group). (**B**) Functional categories significantly affected by age as determined by Gene Ontology Analysis using Seqmonk. From a publicly available list of gene sets (*Merico et al., 2010*), significantly different gene ontology terms are shown (Kolmogorov-Smirnov test, p<0.05, average absolute z-score >1, multiple

*Figure 4 continued on next page*

*Figure 4 continued*

testing correction). Bars are labelled with the number of genes in each set (inside) and the adjusted p-value (outside). (**C**) Average RPKM (read per kilobase million) expression of IFN-I-responsive genes in GFP⁺ CD11b⁺ cDC2s as determined by RNA sequencing. RPKM expression values of the same IFN-I stimulated gene in cDC2s from 2-month-old and 23-month-old mice are connected by lines. (**D-E**) 2–3 month-old and 22–24 month-old C57BL/6 mice were subcutaneously immunised with Eα-GFP in IFA. 22 hr later, *Ifit1* (**D**) and *Mx1* (**E**) mRNA levels were determined in sorted GFP⁺ CD11b⁺ cDC2s by RT-qPCR (n = 4–6 per group/experiment). (**F**) 2–3 month-old and 22–24 month-old C57BL/6 mice were subcutaneously immunised with Eα-GFP in IFA. 6 hr later, *Ifnb1* mRNA levels were determined in cells isolated from the draining lymph nodes (LN) by RT-qPCR (n = 5 per group/experiment). (**G-H**) Representative flow cytometric plot (**G**) and quantitation (**H**) of STAT1 phosphorylation in CD11b⁺ cDC2s (CD11b⁺CD11c⁺MHCII⁺CD172a⁺CD8⁻CD4⁻B220⁻ cells) from 2 to 3 month-old and 22–24 month-old C57BL/6 mice upon ex vivo treatment of LN cells with 50 U murine IFNα for 30 min, or no cytokine controls (n = 4–7 per group/experiment). (**I-J**) *Ifit1* (**I**) and *Mx1* (**J**) mRNA levels were determined in sorted GFP⁺ CD11b⁺ cDC2s by RT-qPCR upon ex vivo treatment of LN cells with 50 U murine IFNα for 3 hr. Expression levels were normalised to age-matched no cytokine controls (n = 5 per group/experiment). (**K-M**) Bone marrow (BM) cells from 3-month-old adult or 23-month-old aged mice were transferred into 2-month-old irradiated CD45.1⁺ C57BL/6 recipient mice. 8 weeks later, these BM chimeras were immunised subcutaneously with Eα-GFP in IFA. 22 hr later, DC populations were analysed by flow cytometry (n = 3–8 per group/experiment). (**K**) Quantitation of GFP⁺CD11b⁺ cDC2 cells in the draining LNs of BM chimeras. (**L-M**) Quantitation of median fluorescence intensity (MFI) levels of CD86 (**L**) and CD80 (**M**) on the surface of GFP⁺CD11b⁺ cDC2s in BM chimeras. (**N**) *Ifnb1* expression in different cell populations FACS-sorted from the draining LNs of unimmunised adult mice or mice immunised with EαGFP 16 hr earlier as determined by RT-qPCR. Cell populations were defined as follows: CD3⁺ T cells, CD19⁺ B cells, macrophages (CD11c⁺CD64⁺F4/80⁺), pDC (CD3⁻CD19⁻B220⁺CD11c⁺PDCA1⁺), cDC1 (CD3⁻CD19⁻CD64⁻F4/80⁻MHC-II⁺CD11c⁺Xcr1⁺) and a population including cDC2s and Langerhans (Lhs) cells (CD3⁻CD19⁻CD64⁻F4/80⁻MHC-II⁺CD11c⁺Xcr1⁻). (**O**) Flow cytometric quantitation of the number of plasmacytoid DCs (pDCs; defined as B220⁺CD11c⁺CD11b⁻PDCA1⁺ cells) in the draining LNs of 2–3 month-old and 22–24 month-old C57BL/6 mice 22 hr after immunisation with Eα-GFP/IFA (n = 5–6 per group/experiment). (**P-S**) Bone marrow (BM) cells from 2-month-old CD45.1⁺ C57BL/6 SJL mice were transferred into 2-month-old or 21-month-old irradiated C57BL/6 recipient mice. 8 weeks later, these BM chimeras were immunised subcutaneously with Eα-GFP in IFA. 22 hr later, DC populations were analysed by flow cytometry (n = 2–5 per group/experiment). (**P-Q**) Quantitation of CD45.1⁺ pDCs (**P**) and GFP⁺CD11b⁺ cDC2s (**Q**) in the draining LNs of BM chimeras. (**R-S**) Quantitation of median fluorescence intensity (MFI) levels of CD86 (**R**) and CD80 (**S**) on the surface of CD45.1⁺GFP⁺CD11b⁺cDC2s in BM chimeras. Bar graphs show the pooled results from at least two experiments except in (**I-J**), which show the results of one of two experimental repeats. Bar heights correspond to the median, and each circle represents one biological replicate. In (**N**) bar heights correspond to the mean and error bars represent standard deviation. *P*-values were determined using the Mann-Whitney test.

The online version of this article includes the following source data for figure 4:

**Source data 1.** Reduced type I interferon (IFN-I) signalling in cDC2s from aged mice.

Many different cell types can produce IFN-I upon infection or immunisation (*Swiecki and Colonna, 2011*; *Ali et al., 2019*). To determine which cells express IFN-I in the draining LN after immunisation we performed RT-qPCR for *Ifnb1* in different FACS-sorted LN cell populations. This revealed that plasmacytoid dendritic cells (pDCs) and CD64⁺F4/80⁺ macrophages express the highest levels of *Ifnb1* 16 hr after immunisation (*Figure 4N*). This indicates a potential link between age-associated defects in cDC2s and the reduced production of *Ifnb1* by pDCs and macrophages.

Previous reports have described an age-associated decline in IFN-I production by pDCs in both humans and mice (*Stout-Delgado et al., 2008*; *Panda et al., 2010*; *Sridharan et al., 2011*; *Agrawal, 2013*; *Agrawal et al., 2017*). In line with this, we found that age-associated defects in cDC2 activation coincided with reduced numbers of pDCs in aged, 22–24 month-old mice compared to adult controls upon immunisation (*Figure 4O*). To understand whether the reduction in pDCs was due to the aged microenvironment, we irradiated aged mice and reconstituted these mice with BM cells from 2 to 3 month-old animals. This revealed that the reduced number of pDCs in ageing is not an intrinsic feature of aged stem cells, but is driven by age-associated changes in their microenvironment (*Figure 4P*). In these chimeras GFP⁺ CD11b⁺cDC2 cell numbers and the expression of CD80/86 on GFP⁺ CD11b⁺ cDC2 cells derived from young stem cells were reduced in aged BM recipients (*Figure 4Q–S*). Together, this suggests that the poor activation of cDC2s in the LNs of aged animals could be driven by reduced numbers of IFN-I-producing pDCs due to age-related changes in their microenvironment.

## Older persons have impaired IFN-I responses after vaccination

As observed in mice, the expression of IFN-I inducible genes is increased in the blood of people in the first few days following vaccination (*Nakaya et al., 2011*; *Athale et al., 2017*). Using data from Henn and colleagues (*Henn et al., 2013*), an early IFN-I gene signature was identified after seasonal influenza vaccination in humans. This response peaks in the blood one day after vaccination

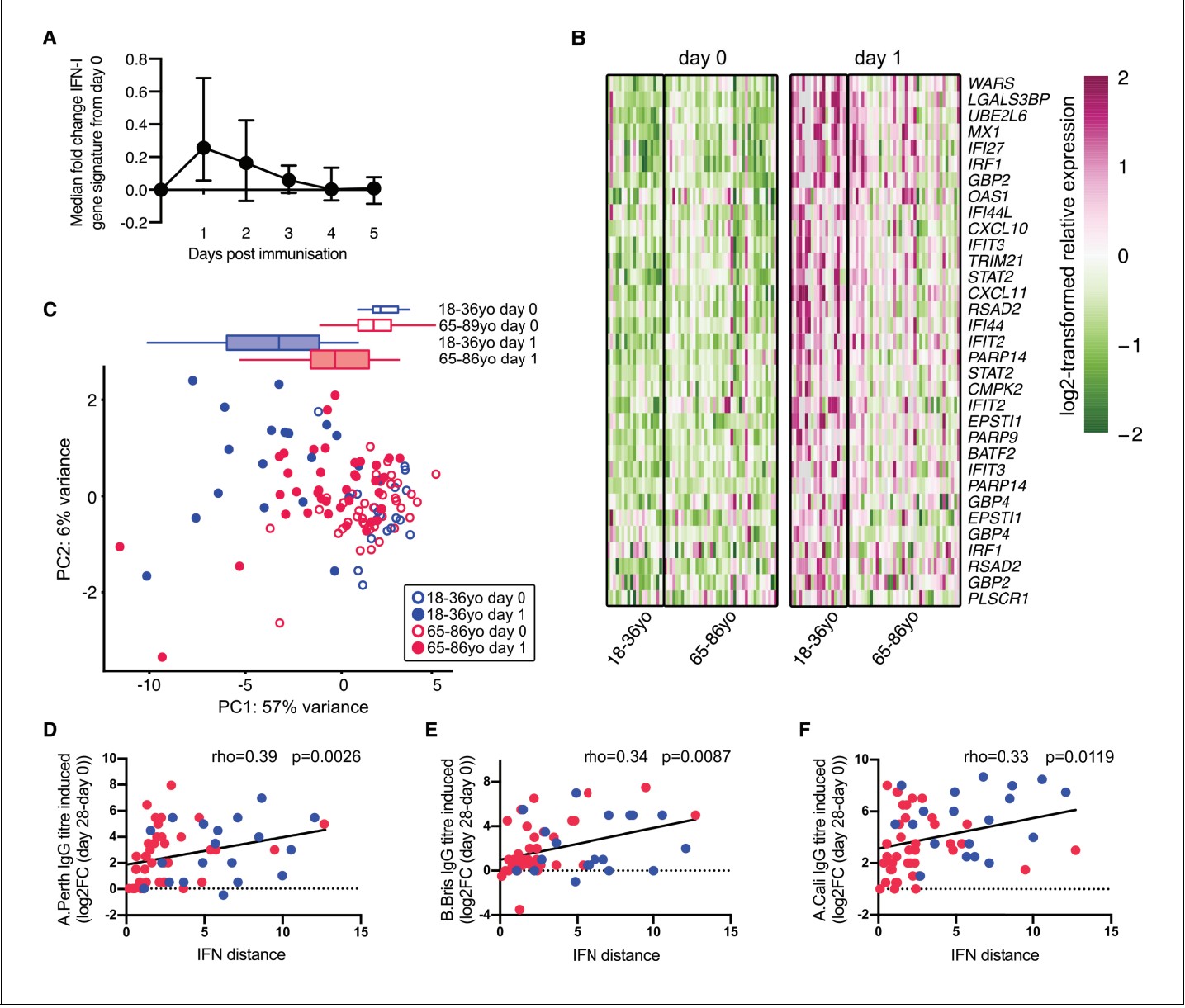

**Figure 5.** Poor vaccine responses in older persons correlate with attenuated IFN-I signalling. (**A**) Time course of the expression of IFN-I-stimulated genes in whole PBMC after influenza vaccination. Median expression of IFN-I-stimulated genes were calculated for each sample at each day and normalised by the day 0 value. The mean and range for five individuals are shown. (**B**) Heatmap of expression values of a curated list of significant probe sets for IFN-I-stimulated genes (ISGs) induced with a log2-fold change of >0.5 determined using the dataset in (**A**), and applied to days 0 and 1 in 18–36 year-old (18–36 yo; n = 19) or 65–86 year-old (65–85 yo; n = 39) individuals. (**C**) Principal component analysis (PCA) of the differentially expressed ISGs from (**B**) with data from younger and older individuals plotted in blue and pink, respectively. Open circles represent data from day 0, closed circles represent data from day 1 after vaccination, and boxplots represent the distribution of PC1 coordinates for each group. (**D-F**) The distance from baseline to day 1 after vaccination on the PCA plot in (**C**) ('IFN-I distance') correlates with log2 fold-changes (day 28/day 0) of HAI titres for A/Perth 2009 (**D**), B/Brisbane 2008 (**E**) and A/California 2009 (**F**). In (**F-H**), Spearman's correlation coefficients (rho) and their *p*-values are shown and younger and older individuals are plotted in blue or pink, respectively. Data are reanalysed from publicly available datasets (*Franco et al., 2013*; *Henn et al., 2013*; *Nakaya et al., 2015*).

The online version of this article includes the following source data for figure 5:

**Source data 1.** Poor vaccine responses in older persons correlate with attenuated IFN-I signalling.

(*Figure 5A*). We further curated this list of IFN-I responsive genes to identify those that are robustly induced in human blood one day after vaccination. Application of this curated IFN-I vaccination

signature to another dataset (*Franco et al., 2013*; *Nakaya et al., 2015*) showed that these genes were induced in older persons after influenza vaccination, but to a lesser extent than in younger people (*Figure 5B*), similar to our findings in aged mice (*Figure 4*). We performed PCA of the curated vaccine-induced IFN-I genes to determine how much an individual's IFN-I gene signature had

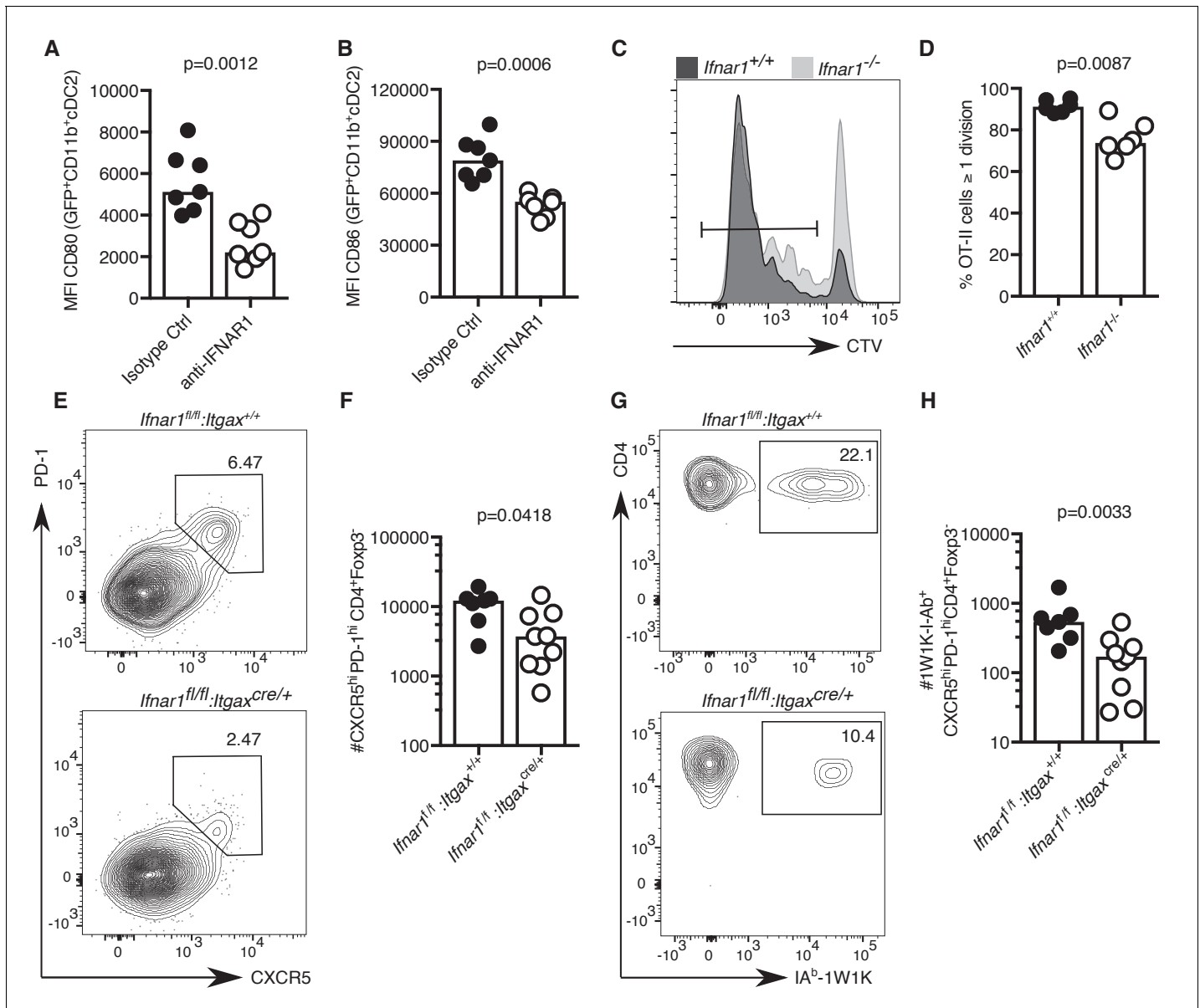

**Figure 6.** Lack of IFN-I signalling in DCs impairs Tfh cell differentiation. (A–B) Quantitation of median fluorescence intensity (MFI) levels of CD80 and CD86 on GFP+ CD11b+ cDC2s in mice treated with anti-IFNAR1 blocking or isotype control antibodies (n = 7 per group/experiment). (C-D) Representative flow cytometric histograms (C) and quantitation (D) of divided CTV+CD45.1+CD4+, which had been transferred into *Ifnar1*−/− and *Ifnar1*+/+ mice and immunised with ovalbumin (OVA) in Alum three days earlier (n = 6 per group/experiment). (E-H) Flow cytometric analysis (E, G) and quantitation (F, H) of total (E-F) and 1W1K-I-Ab+ (G-H) CXCR5hiPD-1hiFoxp3-CD4+ T follicular helper (Tfh) cells isolated from *Ifnar1*fl/fl:*Itgax*cre/+ or *Ifnar1*fl/fl:*Itgax*+/+ control mice seven days after immunisation with NP-1W1K in Alum (n = 8–9 per group/experiment). Bar graphs show the results of one of two experimental repeats. Bar height corresponds to the median, and each circle represents one biological replicate. *P*-values were determined using the Mann-Whitney test. Supporting data is shown in *Figure 6—figure supplement 1*.

The online version of this article includes the following source data and figure supplement(s) for figure 6:

**Source data 1.** Lack of IFN-I signalling in DCs impairs Tfh cell differentiation.

**Figure supplement 1.** Lack of IFN-I signalling in DCs does not affect early antigen-specific antibody responses.

**Figure supplement 1—source data 1.** Lack of IFN-I signalling in DCs does not affect early antigen-specific antibody responses.

changed one day after vaccination compared to the baseline expression before vaccination ('IFN-I distance') (*Figure 5C*). Day one samples from young people clustered distinctly from their pre-vaccination samples, whereas older persons showed a diminished early IFN-I response, as shown by a shorter IFN distance between day 0 and day 1 samples on the PCA plot. The IFN-I distance was reduced by half in older compared to younger individuals (p<0.001). Increased IFN distances correlated positively with an increased neutralising antibody titre 28 days after vaccination (*Figure 5D–F*), demonstrating a correlation between the early IFN-I response and antibody titres upon vaccination in people.

## Lack of IFN-I signalling in DCs results in impaired Tfh cell formation

IFN-I signalling has previously been implicated in Tfh cell priming and protective immune responses to vaccination in mice (*Proietti et al., 2002*; *Cucak et al., 2009*). Hence, we hypothesised that reduced IFN-I signalling in cDC2s of aged mice could be linked with defective T cell priming. Blockade of IFN-I signalling prior to immunisation significantly reduced the expression of the co-stimulatory ligands CD80 and CD86 on cDC2s (*Figure 6A–B*), showing that a reduction of IFN-I signalling can partly recapitulate the ageing cDC2 phenotype. Furthermore, absence of IFNαR in recipient mice results in reduced T cell proliferation of adoptively transferred IFNαR-sufficient ovalbumin-specific OTII cells three days after immunisation (*Figure 6C–D*), similar to what had been observed in aged mice (*Figure 3C–D*). To formally test the link between IFNαR signalling in DCs and Tfh cell formation, *Ifnar1*$^{fl/fl}$:*Itgax*$^{cre/+}$ mice, that lack IFNαR on CD11c$^+$ cells such as DCs, and their IFNαR-sufficient *Ifnar1*$^{fl/fl}$:*Itgax*$^{+/+}$ littermates were immunised with NP-1W1K in Alum. In the absence of IFNAR1 on DCs, there was a defect in Tfh cell formation seven days after immunisation (*Figure 6E–H*), without affecting serum levels of antigen-specific antibody at this early time-point (*Figure 6—figure supplement 1A–C*). This indicates that the lack of IFN-I signalling in DCs can mimic the age-associated impairment of Tfh cell differentiation.

## The TLR7 agonist imiquimod boosts cDC2s and tfh cell numbers

Our data demonstrate that cDC2 and Tfh cells from aged mice have impaired responses to vaccination. In an attempt to boost the response of these cell types upon vaccination we applied a cream containing the TLR7-agonist imiquimod (*Reiter et al., 1994*; *Moore et al., 2001*) to the skin over the immunisation sites (experimental set-up shown in *Figure 7A*). Imiquimod treatment is known to induce IFN-I along with other cytokines and chemokines (*Chen, 1988*; *Bottrel et al., 1999*; *Sauder, 2003*; *Schön and Schön, 2007*). Imiquimod treatment increased expression of the ISGs *Ifit1* and *Mx1* in antigen-bearing cDC2s from both 2–3- and 22–24 month-old mice compared to age-matched no-imiquimod controls one day after immunisation (*Figure 7B–C*; *Figure 7—figure supplement 1A–B*). This was associated with a two- to three-fold increase in the number of total and GFP$^+$ CD11b$^+$ cDC2s in draining LNs (*Figure 7D–E*; *Figure 7—figure supplement 1C–D*) and with a two-fold increase in CD80 and CD86 on their surfaces (*Figure 7F–G*, *Figure 7—figure supplement 1E–F*). Topical imiquimod treatment of *Ifnar1*$^{-/-}$ and *Ifnar1*$^{+/+}$ mice revealed that the enhancing effects of imiquimod on cDC2 numbers and activation were largely, but not completely, dependent on IFN-I signalling (*Figure 7H–J*). This demonstrates that imiquimod treatment can boost the reduced IFN-I response, and revert the numerical and co-stimulatory defects observed in cDC2s from aged mice.

Topical imiquimod treatment potently enhanced total and 1W1K-specific Tfh cell numbers in both 2–3- and 22–24 month-old mice seven days after immunisation (*Figure 8A–F*). This boost in Tfh cell numbers was not dependent on intact IFN-I signalling in CD11c$^+$ cells, as topical imiquimod treatment enhanced total and 1W1K-specific Tfh cell numbers in *Ifnar1*$^{fl/fl}$:*Itgax*$^{cre/+}$ mice to the same extent as in *Ifnar1*$^{fl/fl}$:*Itgax*$^{+/+}$ control mice seven days after immunisation (*Figure 8G–H*). That imiquimod can enhance Tfh cells independently of IFN-I signalling in DCs indicates TLR7 stimulation has multi-faceted roles in enhancing Tfh responses. Together, these results show that imiquimod treatment can restore cDC2 activation and Tfh cell differentiation in aged mice. This was linked with a doubling of GC B cell numbers in aged mice, but not in younger animals (*Figure 8I–K*). Imiquimod treatment did not affect serum levels of antigen-specific antibodies at this early time-point when most antibodies come from the extrafollicular plasmablast response (*Figure 8—figure supplement 1A–C*; *MacLennan et al., 2003*).

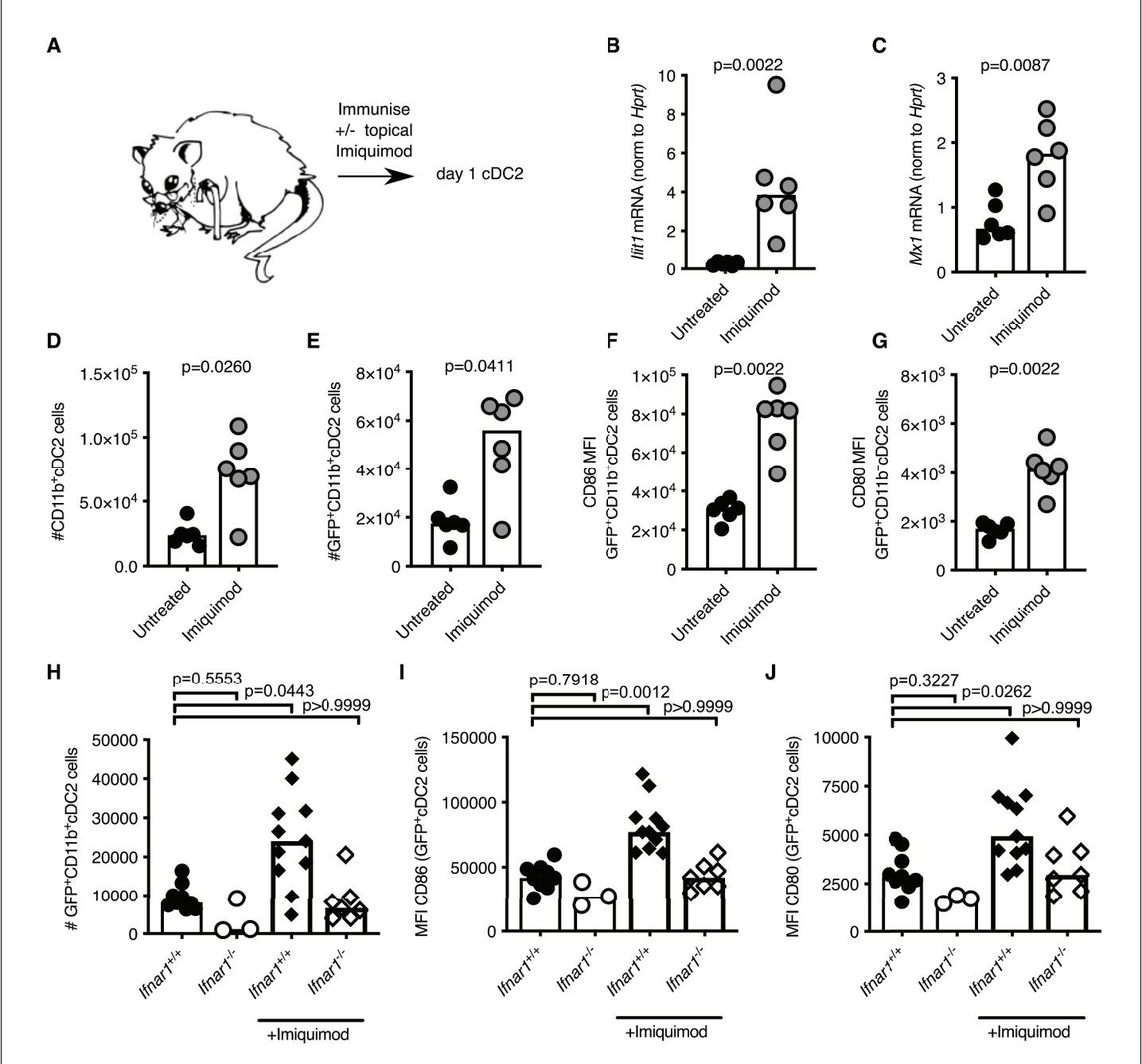

**Figure 7.** Imiquimod rejuvenates cDC2s in aged mice by enhancing IFN-I signalling. (**A**) Schematic representation of the experimental set-up. (**B-G**) 22–24 month-old mice were immunised subcutaneously with Eα-GFP in IFA. Half of the mice were topically treated with imiquimod cream over their immunisation sites. 22 hr after immunisation with Eα-GFP in IFA, *Ifit1* (**B**) and *Mx1* (**C**) mRNA expression in sorted GFP⁺ CD11b⁺ cDC2s was analysed by RT-qPCR. (**D-E**) Flow cytometric quantitation of total (**D**) and GFP⁺(**E**) CD11b⁺ cDC2 cells in the draining lymph nodes (LNs) of 22–24 month-old mice with or without imiquimod treatment. (**F-G**) Quantitation of median fluorescence intensity (MFI) levels of CD86 (**F**) and CD80 (**G**) on the surface of GFP⁺ CD11b⁺ cDC2s in 23-month-old mice with or without imiquimod treatment. (**H-J**) 2 month old *Ifnar1⁻/⁻* and *Ifnar1⁺/⁺* mice were immunised subcutaneously with Eα-GFP in IFA and some of the mice were additionally treated with imiquimod cream over their immunisation sites. (**H**) 22 hr later the number of GFP⁺ CD11b⁺ cDC2 cells in the draining lymph nodes (LNs) were quantified. (**I-J**) Quantitation of median fluorescence intensity (MFI) levels of CD86 (**I**) and CD80 (**J**) on the surface of these GFP⁺ CD11b⁺ cDC2s. Bar graphs show the results of one of two independent experiments (**B-G**; n = 6 per group/experiment) or the pooled results from two experiments (**H-J**; n = 3–11 per group). Bar height corresponds to the median, and each circle represents one biological replicate. In (**B-G**) *p*-values were determined using the Mann-Whitney test. In (**H-J**) *p*-values were determined by comparing each group to the '*Ifnar1⁺/⁺* no imiquimod control' group using the Kruskal Wallis test with Dunn's multiple testing correction. Supporting data is shown in *Figure 7—figure supplement 1*.

*Figure 7 continued on next page*

*Figure 7 continued*

The online version of this article includes the following source data and figure supplement(s) for figure 7:

**Source data 1.** Imiquimod rejuvenates cDC2s in aged mice by enhancing IFN-I signalling.
**Figure supplement 1.** Imiquimod induces IFN-I signalling and boosts cDC2 responses.
**Figure supplement 1—source data 1.** Imiquimod boosts Tfh cell differentiation in young mice.

## Discussion

The humoral immune response to vaccination is diminished in older individuals, at least in part due to defects in the GC response (*Linterman, 2014*; *Gustafson et al., 2018*). Here, we show that the formation of Tfh cells is impaired in older persons and aged mice. In mice, we show topical application of the TLR7 agonist imiquimod corrected the age-dependent defects in cDC2s and increased the formation of Tfh and GC B cells in aged mice upon immunisation. This demonstrates that the defect in the Tfh cell response in aged individuals is not irreversible, and can be corrected by exogenous stimuli. Supporting our data, a clinical trial has demonstrated that topical imiquimod treatment at the time of seasonal influenza vaccination enhances antibody responses to the vaccine in older persons (*Hung et al., 2014*; *Hung et al., 2016*). This, together with the data presented here, demonstrates that changing vaccination approaches is a rational strategy for improving vaccine efficacy in older persons, a challenge of increasing importance with the rising population age world-wide.

Ageing has been reported to impair DC activation (*Agrawal et al., 2007*; *Moretto et al., 2008*), the production of IFN-I cytokines by pDCs in both humans and mice (*Stout-Delgado et al., 2008*; *Panda et al., 2010*; *Sridharan et al., 2011*; *Agrawal, 2013*; *Agrawal et al., 2017*), and the formation of Tfh cells (*Garcia and Miller, 2001*; *Eaton et al., 2004*; *Lefebvre et al., 2012*; *Linterman, 2014*; *Gustafson et al., 2018*; *Nikolich-Žugich, 2018*). In addition, Brahmakshatriya and colleagues have demonstrated that transferring activated, in vitro BM-derived DCs into aged mice can boost both the GC and Tfh cell response upon immunisation (*Brahmakshatriya et al., 2017*). Our study demonstrates that age-associated defects in the early induction of IFN-I expression, probably by pDCs and macrophages, results in impaired expression of co-stimulatory molecules on cDC2s. The impact of reduced IFN-I stimulation on cDC2s in aged mice can be multifactorial: in DCs, IFN-Is promote the expression of co-stimulatory molecules and the cytokines IL-6, IL-27 and IL-1β (*Luft, 1998*; *Moretto et al., 2008*; *Cucak et al., 2009*; *Batten et al., 2010*; *Gringhuis et al., 2014*; *Hassanzadeh-Kiabi et al., 2017*). These molecules have been shown to enhance the expression of Bcl6, CXCR5 and ICOS by $CD4^+$ T cells, thereby supporting differentiation towards the Tfh cell phenotype (*Cucak et al., 2009*; *Barbet et al., 2018*). In particular, IL-6 production is reduced in DCs from aged mice (*Brahmakshatriya et al., 2017*), which may act together with reduced co-stimulation to contribute to poor Tfh cell priming in aged mice.

In an attempt to enhance cDC2 and Tfh cell responses upon vaccination, we applied a cream containing the TLR7-agonist imiquimod, which has been shown to induce IFN-Is (*Chen, 1988*; *Bottrel et al., 1999*; *Sauder, 2003*) to the skin of mice upon immunisation. Treatment with the TLR7 agonist imiquimod at the time of immunisation restored IFN-I signalling in cDC2s in aged mice, increased their numbers in the draining lymph node and enhanced CD80/CD86 expression on their surface. This demonstrated that IFN-I is an important signalling pathway to target to improve cDC2 functions in ageing. However, IFN-I signalling in cDC2s was not uniquely required for the imiquimod-mediated increase in Tfh cell numbers in young mice. This suggests that other, non-IFN driven effects of imiquimod, such as IL-6 production, can support Tfh cell formation (*Schön and Schön, 2007*; *Walter et al., 2013*). Alternatively, it may be that the induction of IFN-I signalling in cells other than DCs, such as B cells or Tfh cells themselves is sufficient to promote Tfh cells (*Le Bon et al., 2006*; *Hervas-Stubbs, 2011*; *Nakayamada et al., 2014*; *Riteau et al., 2016*; *Li et al., 2018*). Together, this indicates that the TLR7 agonist imiquimod can boost cDC2 and Tfh cells using more than one mechanism, reinforcing its potential as a vaccine adjuvant.

In our experiments, imiquimod treatment rescued the age-dependent defects in cDC2s and Tfh cell differentiation but did not fully restore GC B cell numbers or early antigen-specific antibody responses in aged mice to levels observed in younger adult mice. This could be due to additional age-associated changes in other cell types involved in the GC and extrafollicular antibody response (*Linterman, 2014*; *Gustafson et al., 2018*; *Nikolich-Žugich, 2018*). The most obvious hypothesis to

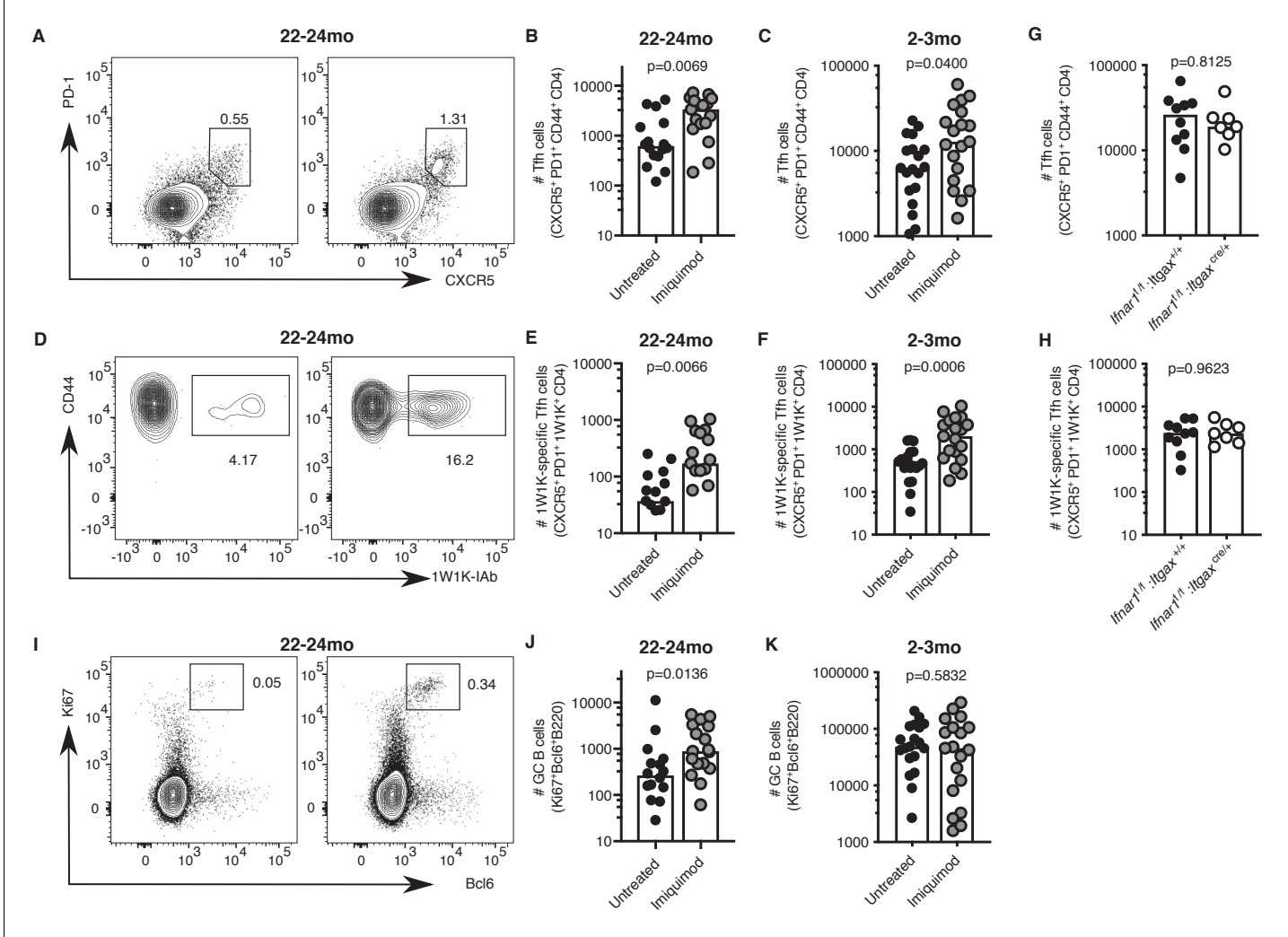

**Figure 8.** Imiquimod rejuvenates Tfh cell differentiation. 2–3 month-old and 22–24 month-old C57BL/6 mice (A-F, I-K) or 2 month old *Ifnar1^{fl/fl}:Itgax^{cre/+}* and *Ifnar1^{fl/fl}:Itgax^{+/+}* littermate controls (G-H) were all subcutaneously immunised with NP-1W1K in Alum and then either topically treated with imiquimod cream over their immunisation sites ('Imiquimod', *Ifnar1^{fl/fl}:Itgax^{cre/+}* and *Ifnar1^{fl/fl}:Itgax^{+/+}* mice) or left untreated ('Untreated'). Seven days later, draining LN cells were analysed by flow cytometry. (A-F) Representative flow cytometric plots and quantitation of CXCR5^{hi}PD-1^{hi}CD4^{+} T follicular helper (Tfh) cells (A-C) as well as antigen-specific 1W1K-I-Ab^{+} Tfh cells (D-F) in 22–24 month-old (A-B, D-E) and 2–3 month-old (C, F) mice. (G-H) Quantitation of CXCR5^{hi}PD-1^{hi}CD4^{+} T follicular helper (Tfh) cells (G) as well as antigen-specific 1W1K-I-Ab^{+} Tfh cells (H) in two month old *Ifnar1^{fl/fl}:Itgax^{cre/+}* and *Ifnar1^{fl/fl}:Itgax^{+/+}* littermate controls seven days after immunisation with NP-1W1K in Alum and imiquimod treatment. (I-K) Representative flow cytometric plots and quantitation of B220^{+}Ki67^{+}Bcl6^{+}germinal centre (GC) B cells in 22–24 month-old (I-J) and 2–3 month-old (K) mice seven days after immunisation with NP-1W1K in Alum with or without topical imiquimod treatment. Bar graphs show the results of one of three independent experiments (G-H; n = 6–10 per group/experiment), or the pooled results of three independent experiments (A-F, I-K; n = 4–7 per group/experiment). Bar height corresponds to the median, and each circle represents one biological replicate. *P*-values were determined using the Mann-Whitney test. Supporting data is shown in *Figure 8—figure supplement 1*.

The online version of this article includes the following source data and figure supplement(s) for figure 8:

**Source data 1.** Imiquimod rejuvenates Tfh cell differentiation.

**Figure supplement 1.** Imiquimod treatment does not affect early antigen-specific antibody responses.

**Figure supplement 1—source data 1.** Imiquimod treatment does not affect early antigen-specific antibody responses.

explain a poor GC response is that ageing results in cell-intrinsic defects in the GC B cells themselves. However, adoptive transfers of aged B cells into young hosts show that B cells from aged mice are capable of forming GCs in a young environment, suggesting that a B cell-intrinsic defect is not the cause of the poor GC response in aged mice (*Yang et al., 1996*). These data therefore implicate B cell-extrinsic factors as additional contributors to the poor GC response in ageing. It is

noteworthy that both T follicular regulatory (Tfr) cells and FDCs have been linked to the age-dependent diminution of the GC response. The GC response is negatively regulated by Tfr cells (*Stebegg et al., 2018*), that are reported to be increased in number in aged mice and this overrepresentation of Tfr cells may result in excessive suppression of the GC response in older animals (*Sage et al., 2015*). There is also evidence that FDCs, stromal cells which are essential for the maintenance of the GC, are impaired in ageing (*Wang et al., 2011*). FDCs in aged mice form smaller networks and present fewer antigen-containing immune complexes on their surfaces after immunisation (*Aydar et al., 2003*; *Turner and Mabbott, 2017*). This is likely to affect the ability of B cells to capture antigen for presentation to Tfh cells, which in turn provide B cell growth and differentiation cues. This suggests that the age-associated defect in GC B cell expansion in mice is linked not only with a defect in T cell priming but also with other factors such as reduced antigen retention on FDCs and increased suppression by Tfr cells.

Several strategies are currently being used to enhance the response to vaccination in older persons, including modifications of adjuvants (*Frech et al., 2005*) or administration of increased antigen doses (*Remarque et al., 1993*). Hung and co-workers have shown that topical imiquimod treatment at the time of vaccination enhances the antibody responses to influenza vaccination in both younger and older persons (*Hung et al., 2014*; *Hung et al., 2016*). We have previously shown that the poor gut GC response in aged mice can be boosted by replenishing the gut microbiome with that of a younger animal (*Stebegg et al., 2019*). Together with the data presented here, this demonstrates that age-related defects in the GC response are not irreversible and can be targeted therapeutically to improve immune responses in older individuals. Because imiquimod can correct defective IFN-I signalling and the associated cellular defects in cDC2s, and can also boost Tfh cell formation in aged animals, this compound could play a key role in improving T-dependent vaccine responses, especially in the older members of our communities.

## Materials and methods

### Human seasonal influenza vaccination cohort

The main research objective of the human seasonal influenza vaccination study was to characterise the cTfh and antibody response against seasonal influenza vaccination in humans in different age groups. To that end, peripheral blood was tested from 34 healthy UK adults (18–75 years of age), who were vaccinated with the trivalent influenza vaccine (2014/2015). Researchers remained blinded to sample age throughout sample processing and data acquisition, and samples were selected at random to be thawed and stained (2–10 samples per experiment). All human blood and tissue was collected in accordance with the latest revision of the Declaration of Helsinki and the Guidelines for Good Clinical Practice (ICH-GCP). The seasonal UK influenza vaccination cohort was collected with UK local research ethics committee approval (REC reference 14/SC/1077), using the facilities of the Cambridge Bioresource (REC reference 04/Q0108/44). Written informed consent was received from all volunteers.

Peripheral blood was tested from 34 healthy UK adults (18–36 years of age n = 16, 65–75 years of age n = 18) who were vaccinated with the trivalent influenza vaccine (northern hemisphere winter 2014–15). Blood samples were collected into silica-coated tubes (for serum) and EDTA-coated tubes (for cells) on the day of vaccination (prior to administration of the vaccine), day 7 and day 42 after vaccination, with approval from the UK local research ethics committee (REC reference 14/SC/1077), and using the facilities of the Cambridge Bioresource (REC reference 04/Q0108/44). Peripheral blood mononuclear cells (PBMCs) were isolated by density centrifugation on Histopaque-1077 (Sigma), and then cryopreserved in freezing medium (10% dimethyl sulfoxide, 90% FCS, both Sigma) and kept in liquid nitrogen prior to analysis by flow cytometry. Cryopreserved PBMC were thawed and rested in complete medium (RPMI-1640 with 10% FCS, 100 U/ml penicillin and 100 μg/ml streptomycin, all Invitrogen) for 1 hr at 37°C. Cells were resuspended at $4 \times 10^7$ per ml, then Fc receptors were blocked using human IgG (Sigma), followed by staining with antibodies outlined in *Table 1* and analysis on a BD Aria Fusion cell sorter. A dump channel consisting of viability dye and antibodies to CD14, CD16, and CD19 was used to exclude unwanted cell types from the analysis of circulating Tfh cells (CD3[+]CD4[+]CD45RA[-]CXCR5[+]PD1[+++]), as previously described (*Hill et al., 2019*). The IgG response was measured by Luminex using magnetic beads coated with full length recombinant

**Table 1.** Antibodies used for flow cytometry of human PBMC.

| Antibody | Company and clone | Dilution |
|---|---|---|
| eFluor780 Viability dye | eBioscience | 1:5000 |
| APC-eFluor780-coupled anti-human CD14 | eBioscience (61D3) | 1:50 |
| APC-eFluor780-coupled anti-human CD16 | eBioscience (eBioCB16) | 1:50 |
| APC-eFluor780-coupled anti-human CD19 | eBioscience (HIB19) | 1:50 |
| BUV395-coupled anti-human CD3 | BD (UCHT1) | 1:100 |
| PerCp-Cy5.5-coupled anti-human CD4 | BD (RPA-T4) | 1:50 |
| BUV737-coupled anti-human CD45RA | BD (HI100) | 1:25 |
| PE-Cy7-coupled anti-human PD1 | eBioscience (eBioJ105) | 1:25 |
| BB515-coupled anti-human CXCR5 | BD (RF8B2) | 1:25 |

haemagglutinin proteins from influenza strain A/Texas/50/2012 (A/Tex12), as previously reported (*Wang et al., 2015b*).

## Mouse housing and husbandry

C57BL/6, Ifnar1$^{-/-}$ (*Skarnes et al., 2011*), Ifnar1$^{flox/flox}$ (*Le Bon et al., 2006*),: Itgax$^{cre/+}$ (*Caton et al., 2007*), OTII TCR-Tg (*Barnden et al., 1998*) and TCR7 TCR-Tg (*Neighbors et al., 2006*) mice were bred and maintained in the Babraham Institute Biological Support Unit (BSU), where C57BL/6Babr mice were also aged. No primary pathogens or additional agents listed in the FELASA recommendations (*Mähler et al., 2014*) were detected during health monitoring surveys of the stock holding rooms. Ambient temperature was ~19–21°C and relative humidity 52%. Lighting was provided on a 12 hr light: 12 hr dark cycle including 15 min 'dawn' and 'dusk' periods of subdued lighting. After weaning, mice were transferred to individually ventilated cages with 1–5 mice per cage. Mice were fed CRM (P) VP diet (Special Diet Services) ad libitum and received seeds (*e.g.* sunflower, millet) at the time of cage-cleaning as part of their environmental enrichment. All mouse experimentation was approved by the Babraham Institute Animal Welfare and Ethical Review Body. Animal husbandry and experimentation complied with existing European Union and United Kingdom Home Office legislation and local standards (PPL: P4D4AF812). Young mice were 6–14 weeks old, and aged C57BL/6 mice 90–105 weeks old when experiments were started. All experimental mice were housed in the same room. H2$^{+/-}$ mice (*Madsen et al., 1999*) and their wildtype littermates were maintained in specific pathogen–free facilities at the University of Leuven. All experiments were approved by the University of Leuven ethics committee.

Mice were randomly allocated into age- and sex-matched experimental groups by staff of the Babraham Institute Experimental Support Unit or on the online mouse colony management system. Blinding was not always possible due to visible phenotypic differences between aged and young mice. For ageing experiments, a minimum of 5 mice per group were chosen based on previous experience in the lab comparing GC responses in young and aged mice. Limited availability of aged mice was a restricting factor in our study design. A minimum of 3 mice per group were chosen for mechanistic studies, based on a literature-based assessment of the expected effect size between study groups. Due to limited availability of aged female mice, all experiments with aged mice were conducted with males. Other experiments were conducted with both male and female mice. All experiments were repeated 2–4 times. Significant changes were reproducible between experimental repeats. Some of the aged mice carried lymphomas or solid tumours which affected their immune system, these mice were excluded from the analysis based on the following criteria: Mice with visible lymphomas or large expansions of Ki67$^+$ proliferating lymphocytes (as determined by Tukey's outlier test) were excluded. These criteria were in place prior to initiation of the study, as part of normal practice for the use of ageing mice at the Babraham Institute.

## Subcutaneous immunisations

Mice were immunised with either 1W1K-conjugate, OVA (ovalbumin), HEL (Hen Egg Lysozyme), NP-OVA (4-hydroxy-3-nitrophenylacetyl-ovalbumin) or NP-KLH (4-hydroxy-3-nitrophenylacetyl-KLH) in

Alum, or Eα-GFP in Incomplete Freund's Adjuvant (IFA). IFA (#F5506), HEL (Lysozyme from chicken egg white, #62970) and OVA (Albumin from chicken egg white; #A5503) were purchased from Sigma-Aldrich, Imject Alum (#77161) was purchased from Thermo Fisher Scientific. NP-KLH (#N-5060–25) and NP-OVA (#N-5051–100) were purchased from Biosearch Technologies. Eα-GFP fusion protein was isolated in-house from XL-1 blue *E. coli* carrying the pTRCHis-Eα-GFP vector using a protocol adapted from *Rush and Brewer (2010)*. Briefly, *E. coli* carrying the pTRCHis-Eα-GFP vector were plated from glycerol stock onto LB/ampicillin agar and incubated overnight at 37 °C. The next day, a single colony was transferred into 5 ml LB and incubated at 37 °C while shaking. The next day, these 5 ml were used to inoculate 1L LB. When the culture was growing in log phase, IPTG was added to a final volume of 1 mM. The culture was left to shake overnight at 37 °C, then the bacteria were pelleted at 5000 g for 15 min. After discarding the supernatant, the bacterial pellet was resuspended in 20 ml lysis buffer (50 mM $NaH_2PO_4$, 300 mM NaCl, 10 mM Imidazole at pH 8.0) by vortexing and incubated on ice for 15 min in ice. After five repeated sonication steps on ice at 30 W for 60 s, the lysate was cleared by centrifugation at 10,000 g for 30 min. This step was repeated until all Eα-GFP was released and the bacterial pellet did not appear green anymore. The clear, green lysate was filtered first through a 0.45 µm syringe filter, then through a 0.22 µm syringe filter. Next, 200 ml of lysate were mixed with 4 ml of Ni-NTA agarose (QIAGEN #30210) and incubated at 4 °C. After one hour, the lysate/agarose mix was loaded onto 5 ml columns (QIAGEN #34964) and left to set. The columns were then washed twice with 25 ml of wash buffer (50 mM $NaH_2PO_4$, 300 mM NaCl, 20 mM Imidazole at pH 8.0). The protein was eluted by four repeated additions of 2 ml elution buffer (50 mM $NaH_2PO_4$, 300 mM NaCl, 250 mM Imidazole at pH 8.0) to the column. The eluate was dialysed against PBS overnight in a D-Tube Dialyzer Mega 3.5 kDa tube (Millipore # 71743–4) at 4 °C. On the next day, the eluate was concentrated using Centriprep centrifugal filters with an Ultracel 10K membrane (Millipore #4304) by centrifugation at 3000 g for 30 min. The concentrated protein was collected, filtered through a 0.22 µm syringe filter and stored at −80 °C until use.

The 1W1K-conjugate was generated from NP-e-Aminocaproyl-OSu (Biosearch Technologies, #N-1021–100), which was first conjugated to streptavidin (Sigma #S4762-10MG). The product was then conjugated at a 1:6 molar ratio of NP-SA to 1W1K-biotin (custom-made by Cambridge Research Biochemicals 'biotin-GSGEA-W-GALANKA-V-DKA-acid') for one hour at room temperature. Unbound peptide was removed by dialysis using Centriprep centrifugal filters with an Ultracel10K membrane (Sigma #4304). NP-1W1K was freshly conjugated for each experiment.

Purified 1W1K-conjugate, NP-KLH, HEL and OVA were first diluted in PBS, then the same volume of Alum was added dropwise to the solution while shaking until a final concentration of 50 µg/ml HEL, 500 µg/ml OVA, 500 µg/ml NP-OVA, 500 µg/ml NP-KLH or 330–500 µg/ml 1W1K-conjugate was reached. After 30 min of vortexing, 100 µl of the emulsion were injected subcutaneously (s.c.) into the hind flanks of recipient mice. 1 mg/ml Eα-GFP was emulsified in IFA by trituration through a 20 g needle. 200 µl of this emulsion were injected subcutaneously into the hind flanks of recipient mice. Mice were euthanised at different time-points after immunisation, as indicated in the main text or figure legends, when draining inguinal LNs were collected.

Where indicated, mice were treated with IFNAR1-blocking antibodies (BioXCell #BE0241), CTLA4-Ig (Orencia Abatacept) or Aldara cream containing 5% imiquimod (MEDA Pharma) prior to, or at the time of immunisation. IFNAR1 blocking was achieved by intraperitoneal injection of 0.75 mg of anti-IFNAR1 blocking antibody or the appropriate isotype control (BioXCell #BE0083) in 200 µl PBS 20 hr before immunisation with Eα-GFP. To block CD28 co-stimulation, 10–100 µg of Abatacept were injected intraperitoneally immediately prior to immunisations with HEL in Alum. For Aldara treatment, mice were shaved on their backs 1–3 days before subcutaneous immunisations into their hind flanks. Directly after immunisation, 50–125 mg of Aldara were applied topically to the shaved backs of the anaesthetised animals. The cream was left to absorb for five minutes before the mice were returned to their cages. Imiquimod-treated and untreated control mice were housed in separate cages to avoid cross-contamination by grooming.

For pertussis toxin experiments, mice were subcutaneously injected into their hind flanks with 100 µl of 5 µg/ml pertussis toxin (Sigma #P7208) in PBS on days 0 and 1. On day 2, they received subcutaneous immunisations with 200 µl of 1 mg/ml Eα-GFP combined with 2.5 µg/ml pertussis toxin in IFA. 22 hr later, mice were euthanised and their draining inguinal LNs were collected for flow cytometry. Control mice were injected with 100 µl PBS on day 0 and 1 and were immunised with Eα-GFP in IFA on day 2.

## Adoptive T cell transfers

To perform adoptive HEL- or OVA-specific T cell transfers, total lymphocytes were isolated from the spleen and peripheral LNs (brachial, axial, superficial cervical, inguinal and mesenteric LNs) of TCR7 or OTII transgenic mice, respectively. These mice also expressed CD45.1$^+$. All cells were stained using the CellTrace Violet (CTV) Cell Proliferation Kit (Invitrogen #C34557; 1:1000 in PBS) for 15 min at 37 ℃, followed by two washes with PBS containing 2 %FBS at 1800 rpm for 5 min at 4 ℃. The percentage of TCR-transgenic CD4$^+$ CTV$^+$ T cells was determined by flow cytometry, detecting TCR7 T cells using anti-TCRVβ3 antibodies and OTII T cells with anti-TCRVα2 antibodies. An equivalent of 1–5 × 10$^6$ TCR7 T cells or 5 × 10$^5$ OTII T cells was transferred intravenously into C57BL/6, *H2$^{+/-}$* or *Ifnar1$^{-/-}$* recipient mice in 100 µl 2% FBS/PBS. The mice were subsequently immunized subcutaneously into their hind flanks with 100 µl of 5–10 µg HEL or 50 µg OVA in Alum. An equivalent of unstained 5 × 10$^4$ OTII T cells in 100 µl 2% FBS/PBS was transferred intravenously into adult and aged C57BL/6 mice for the assessment of Tfh cell differentiation on day 10 after immunisation with 50 µg NP-OVA in Alum. Where indicated, mice were pre-treated intraperitoneally with 1, 50 or 100 µg CTLA4-Ig (Orencia Abatacept) in 100 µl PBS at the time of T cell transfer to block CD86 and CD80 co-stimulation. After three days, the inguinal LNs of each recipient mouse were harvested and pooled. They were mashed through 70 µm filters to obtain single cells which were then stained for flow cytometry. The antibodies used are listed in *Table 2*.

## Bone marrow chimeras

Recipient mice (2-month-old CD45.1$^+$ C57BL/6 SJL or 2-month-old and 21-month-old C57BL/6 mice) were irradiated with 800–1000 rad in two doses and reconstituted *via* intravenous injection with 2–4 × 10$^6$ BM cells isolated from donor mice (2–3 month-old CD45.1$^+$ C57BL/6 SJL, 3-month-old or 23-month-old C57BL/6 mice). BM chimeras were administered neomycin in their drinking water for the first four weeks after BM transfers and were used for experiments eight weeks after successful reconstitution.

## Flow cytometry

For T and B cell stains a single cell suspension from inguinal LNs was generated by pressing the tissues through a 70 µm mesh in 2% FBS in PBS. Cell numbers and viability were determined using a CASY TT Cell Counter (Roche). 1–3 × 10$^6$ cells were transferred to FACS tubes or 96-well plates for subsequent antibody staining. To stain for 1W1K-specific CD4 T cells, cell suspensions were first pre-treated with Dasatinib (BioVision #1568–100, 1:20,000 in DMEM (Dulbecco's modified eagle medium, Gibco #41965–039) containing 10% FBS, 100 U/ml penicillin and 100 µg/ml streptomycin) for 10 min at 37 ℃. Then, a PE-conjugated MHC class II tetramer containing the 1W1K peptide (obtained through the NIH Tetramer Core Facility; PE-coupled 'I-A(b) EAWGALANKAVDKA') was added to each sample at a final concentration of 1:100 and incubated for 2 hr at room temperature. Cells were stained with LIVE/DEAD Fixable Blue Dead Cell Stain (Invitrogen #L23105; diluted 1:1000 in PBS) and incubated with FcR block for 15 min (anti-mouse CD16/32; eBioscience #14-0161-82; diluted 1:50 in 2% FBS in PBS). Surface antibody stains were performed for 1 hr at 4 ℃ in 100 µl Brilliant Stain Buffer (BD Biosciences #563794). For intranuclear staining, cells were fixed with the eBioscience Foxp3/Transcription Factor Staining Buffer (#00-5323-00). Antibody staining with anti-Foxp3, anti-Ki67 and anti-Bcl6 antibodies was performed for 1–2 hr at 4 ℃ in 1 × Permeabilization buffer (eBioscience #00-8333-56). Samples were acquired on an LSRFortessa (BD Biosciences) with stained UltraComp eBeads Compensation Beads (Invitrogen #01-2222-41) as compensation controls. Flow cytometry data were analysed using FlowJo v10 software (Tree Star). The antibodies used are listed in *Table 2*.

For DC analysis, inguinal LNs were harvested and incubated with 10 mg/ml Collagenase D (Roche #11088866001) in plain RPMI medium (Gibco #11875093) for 15–30 min at 37 ℃, followed by gentle pipetting to disrupt the tissue. Cells were washed with PBS containing 2% FBS, before cell numbers were determined using a CASY TT Cell Counter (Roche). After a wash in PBS, isolated cells were stained with LIVE/DEAD Fixable Blue Dead Cell Stain (Invitrogen #L23105; diluted 1:1000 in PBS) on ice for 10 min. After a second wash, they were blocked with FcR block (anti-mouse CD16/32; eBioscience #14-0161-82; diluted 1:50 in 2% FBS in PBS) for 10–15 min at 4 ℃. Surface antibody stains were performed for 45–60 min at 4 ℃ in Brilliant Stain Buffer (BD Biosciences #563794). Samples

**Table 2.** Antibodies and conjugated probes used for flow cytometry and FACS of mouse tissues.

| Antibody | Supplier (Clone) | Dilution |
|---|---|---|
| PE/PE-Cy7-coupled anti-mouse Bcl6 | BD Biosciences (K112-91) | 1:100 |
| PE-Cy7-coupled anti-mouse CD95 | BD Biosciences (Jo2) | 1:200 |
| BV605-coupled anti-mouse IgG1 | BD Biosciences (A85-1) | 1:100 |
| PE-Cy7/BUV395-coupled anti-mouse CD3 | BD Biosciences (145–2 C11) | 1:300 |
| PE-Cy7/BUV395-coupled anti-mouse CD19 | BD Biosciences (1D3) | 1:200-1:300 |
| PE-Cy7/BUV395-coupled anti-mouse B220 | BD Biosciences (RA3-6B2) | 1:300 |
| BUV395-coupled anti-mouse *CD8a* | BD Biosciences (53–6.7) | 1:200 |
| AF647-coupled anti-mouse CD64 | BD Biosciences (X54-5/71) | 1:200 |
| PE-Cf594-coupled anti-mouse CD11b | BD Biosciences (M1/70) | 1:200 |
| BV786-coupled anti-mouse CD103 | BD Biosciences (M290) | 1:200 |
| PE/BV510-coupled anti-mouse CD86 | BD Biosciences (GL1) | 1:300 |
| APC-AF780-coupled anti-mouse PD1 | eBioscience (J43) | 1:200 |
| APC/Foxp3-coupled anti-mouse Foxp3 | eBioscience (FJK-16S) | 1:100-1:200 |
| AF488/AF700-coupled anti-mouse Ki67 | eBioscience (SolA15) | 1:100 |
| Biotin-coupled anti-mouse Gr1 | eBioscience (RB6-8C5) | 1:200 |
| eF450-coupled anti-mouse CD38 | eBioscience (90) | 1:400 |
| PerCp-Cy5.5-coupled anti-mouse CD172a | eBioscience (P84) | 1:200 |
| eF450-coupled anti-mouse CD24 | eBioscience (M1/69) | 1:500 |
| APC/APC-AF870-coupled anti-mouse CD11c | eBioscience (N418) | 1:200 |
| PE-Cy5/APC-coupled anti-mouse CD80 | eBioscience (16-10A1) | 1:300 |
| PerCp-Cy5.5-coupled anti-mouse CD45.2 | eBioscience (104) | 1:200 |
| AF700-coupled anti-mouse MHC-II | eBioscience (M5/114.12.2) | 1:400 |
| BV421-coupled anti-mouse CXCR5 | Biolegend (L138D7) | 1:100 |
| V500/PE/BV605-coupled anti-mouse CD4 | Biolegend (RM4-5) | 1:400-1:800 |
| BV510/BV785-coupled anti-mouse B220 | Biolegend (RA3-6B2) | 1:200-1:400 |
| PerCp-Cy5.5-coupled anti-mouse CD44 | Biolegend (IM7) | 1:200 |
| AF488-coupled anti-mouse GL7 | Biolegend (GL7) | 1:100 |
| BV605/PE-coupled anti-mouse F4/80 | Biolegend (BM8) | 1:200 |
| BV650-coupled anti-mouse XCR1 | Biolegend (ZET) | 1:200 |
| PE-Cy7-coupled anti-mouse CD40 | Biolegend (3/23) | 1:300 |
| AF700-coupled anti-mouse CD45.1 | Biolegend (A20) | 1:200 |
| PerCp-Cy5.5-coupled anti-mouse PDCA-1 | Biolegend (927) | 1:100 |
| PE-coupled anti-mouse TCR Vβ3 | BD Biosciences (KJ25) | 1:100 |
| APC-coupled anti-mouse TCR Vα2 | eBioscience (B20.1) | 1:100 |
| APC/PE-Cy7/BV510-coupled Streptavidin StrepavidinStreptavidin | Biolegend, eBioscience | 1:600 |
| Biotin-coupled anti-mouse Eα 52–68 peptide | eBioscience (eBioY-Ae) | 1:200 |
| PE-coupled 1W1K-IAb tetramer | NIH Tetramer core facility | 1:100 |
| PE-coupled anti-mouse *pSTAT1* | BD Biosciences Phosflow (pY701) | 1:10 |

were acquired on a LSRFortessa 5 and sorted with a BD FACSAria (both BD Biosciences). Flow data were analysed using FlowJo v10 software (Tree Star). The antibodies used are listed in *Table 2*.

To stain for pSTAT1,~$2 \times 10^6$ cells isolated from the inguinal LNs of naïve mice were seeded into sterile round-bottom 96-well plates in 200 µl complete RPMI (RPMI medium (Gibco #11875093) containing 10% FBS (Sigma #F9665), 100 U/ml penicillin/streptomycin (Thermo Fisher Scientific #15140–122) and 55 µM 2-mercaptoethanol (Thermo Fisher Scientific #21985023)). After a 45 min resting

period, they were treated with 50 U/well recombinant murine IFNα (PBL assay science #12105–1) for 30 min and simultaneously stained with fluorochrome-coupled anti-mouse CD8a, B220 and CD11b antibodies. The cells were then washed and fixed with Cytofix (BD Biosciences #554655) for 30 min, followed by a 30 min fixation and permeabilisation step in ice-cold 90% methanol. After three washes in PBS, the cells were stained with anti-mouse pSTAT1 antibodies as well as anti-mouse CD4, CD11c, MHCII, CD172a antibodies for one hour. Samples were acquired on a LSRFortessa 5 and the flow data were analysed using FlowJo v10 software (Tree Star). The antibodies used are listed in *Table 2*.

## Fluorescence-activated cell sorting (FACS)

For RT-qPCR of GFP$^+$ CD11b$^+$ cDC2, cells from total draining LNs were isolated and stained as described above. 800–4000 GFP$^+$ CD11b$^+$ cDC2 were sorted into PCR tubes containing 20 µl of RLT lysis buffer supplied with the RNeasy Micro Kit (Qiagen #74004) using a BD FACSAria or FACS Aria Fusion. For RNA sequencing, we sorted 800 cells into PCR tubes containing 8.5 µl of 1 × lysis buffer provided with the SMART-Seq v4 Ultra Low Input RNA Kit for Sequencing (Clontech # 634890).

To determine the IFN-I responsiveness of cDC2s ex vivo, ~2×10$^6$ cells isolated from the inguinal LNs of naïve mice were seeded into sterile round-bottom 96-well plates in 200 µl complete RPMI. After a 15 min resting period, they were treated with 50 U/well recombinant murine IFNα (PBL assay science #12105–1) for 3 hr and then stained and sorted CD11b$^+$ cDC2s as described above.

To investigate *Ifnb1* expression of different LN cell subsets we sorted different cell populations based on a gating strategy proposed by *Guilliams et al. (2016)*: T cells (CD3$^+$), B cells (CD19$^+$), macrophages (CD11c$^+$F4/80$^+$CD64$^+$), pDCs (CD3$^-$CD19$^-$CD11c$^+$B220$^+$PDCA1$^+$), cDC1s (CD3$^-$CD19$^-$F4/80$^-$CD64$^-$MHC-II$^+$CD11c$^+$Xcr1$^+$) and a population containing both cDC2s and Langerhans cells (CD3$^-$CD19$^-$F4/80$^-$CD64$^-$MHC-II$^+$CD11c$^+$Xcr1$^-$) from the pooled draining LNs of 4–5 mice 16 hr after immunisation with Eα-GFP in IFA. To sort DCs and macrophages, LN were digested as described above and Miltenyi's pan dendritic cell isolation kit (#130-100-875), LS columns (Miltenyi #130-042-401) and a MidiMACS separator (Miltenyi) were used to magnetically enrichment for CD11c$^+$ cells, by following the manufacturer's instructions. The DC-enriched cell fraction was stained as described above. All cell types were sorted into separate 5 ml tubes containing 300 µl FBS using a BD FACSAria Fusion. RNA for RT-qPCR was isolated from cell pellets using Qiagen's RNeasy Mini or Micro Kit (#74104 and #74004).

## Confocal imaging of germinal centres

Draining inguinal LNs were fixed in periodate-lysine-paraformaldehyde (PLP) containing 1% (v/v) PFA (Sigma #P6148), 0.075 M L-Lysine (Sigma #L5501), 0.37 M Na$_3$PO$_4$ (pH 7.4) (Sigma #342483) and 0.01 M NaIO$_4$ (Sigma #210048), for 4 hr at 4 ˚C. After fixation, the samples were dehydrated in 30% sucrose (Sigma #S0389) overnight, embedded in Optimum Cutting Temperature (OCT) medium (VWR #25608–930) on dry ice and stored at −80 ˚C. The frozen tissues were cut into 10 µm sections using a cryostat (Leica Biosystems) at −20 ˚C and again stored at −80˚C. For antibody stains, the slides were first air-dried and then hydrated in 0.5% Tween 20 in PBS (PBS-T). Slides were blocked in 200 µl blocking buffer (PBS containing 2% BSA and 10% goat serum), then permeabilised with 200 µl PBS containing 2% Triton X (Sigma #X100). After three wash steps in PBS-T, the slides were incubated with 200 µl of a primary antibody mix in PBS-T containing 1% BSA at 4 ˚C overnight. Sections were stained with eF450-conjugated rat anti-mouse Foxp3 (clone FJK16S, Thermo Fisher Scientific; 1:50), hamster anti-mouse CD3ε (clone 500A2, Thermo Fisher Scientific; 1:200), rabbit anti-mouse Ki67 (#15580, Abcam; 1:100) and AF647-conjugated rat anti-mouse IgD (clone 11–26 c.2a, Biolegend; 1:100). The next day, the slides were washed in PBS-T three times, then they were incubated with secondary antibodies in 200 µl PBS-T containing 2% goat serum for 2 hr at room temperature. The secondary antibodies used were AF568-conjugated goat anti-hamster IgG (#A-21112, Life Technologies; 1:500) and AF488-conjugated goat anti-rabbit (#150077, Abcam; 1:400). Hydromount mounting medium (National diagnostics #HS-106) was used to mount slides and coverslips were gently placed on top of the slides. Slides were dried overnight for the mounting medium to set. Images were acquired using a Zeiss 780 microscope using 10×, 20 × and 40 × objectives. Image analysis was performed using ImageJ.

## RNA isolation and quantitative Real-Time PCR (RT-qPCR)

RNA isolation from ex vivo isolated cells was performed using Qiagen's RNeasy Mini or Micro Kit (#74104 and #74004) following the manufacturer's instructions. Homogenisation of the samples was achieved by vortexing for 1 min or by using QIAshredders (Qiagen #79654). RNA concentrations obtained from the RNA isolation were measured using the NanoDrop system (Thermo Fisher Scientific).

The TaqMan Gene Expression Assay (Thermo Fisher Scientific #4331182) for *Ifnb1* (Mm00439552_s1) detects genomic DNA, so RNA samples were treated with the Turbo DNA-free kit (Thermo Fisher Scientific #AM1907) according to the manufacturer's protocol to remove any contaminating genomic DNA for RT-qPCR. cDNA was generated from pre-treated RNA samples using the Quantitect reverse transcription kit (Qiagen #205311) and RT-qPCR for *Ifnb1*, *Mx1* (Mm00487796_m1) and *Ifit1* (Mm00515153_m1) was performed using the Platinum Quantitative PCR SuperMix-UDG (Thermo Fisher Scientific #11730025). In some cases, RT-qPCR using TaqMan Gene Expression Assays for *Mx1* and *Ifit1*, which do not detect genomic DNA, was performed directly on RNA using Thermo Fisher Scientific's TaqMan RNA-to-CT 1-Step Kit (#4392656) following the manufacturer's protocol. All RT-qPCR reactions were assembled in PCR 96-well (Bio-Rad #MLL9601) or 384-well plates (Bio-Rad #HSP3805), adding 2 µl of template RNA or cDNA (10–50 ng per reaction) to 8 µl of a master mix containing the appropriate TaqMan Gene Expression Assay. Expression levels were normalised to *Hprt* (Mm03024075_m1), which was found to be stably expressed under all experimental conditions.

All samples were run in duplicates or triplicates on a BioRad CFX96 or CFX384 Real-Time System. The $2^{-\Delta\Delta Ct}$-method was applied for relative quantification of mRNA levels. Samples from young mice were used as calibrators. As above, Cq values were exported from the CFX Manager software (Bio-rad).

## Enzyme-linked immunosorbent assay (ELISA)

ELISA plates (Thermo Fisher Scientific 96F Maxisorp #456537) were coated overnight at 4 °C with 10 µg/ml NP20-BSA (Biosearch Technologies #N-5050H-100) or 2.5 µg/ml NP7-BSA (Biosearch Technologies #N-5050L-100) in PBS. The next day, plates were washed 4 times in wash buffer containing 0.05% Tween 20 in PBS and blocked with 1% BSA in PBS for 1 hr at room temperature. After an additional wash step, sera were loaded onto the plates. The starting dilution for sera was 1:50–200 in 1% BSA in PBS. This initial dilution was titrated down the plate at a 1:3 or 1:4 ratio. The plates were incubated for 2–3 hr at room temperature and after another wash step 50 µl of polyclonal goat anti-mouse IgG1 HRP-conjugated antibodies (Abcam #ab97240; 1:10,000 in PBS) were added for 2 hr at room temperature. The plates were developed with 100 µl/well TMB (Biolegend #421101) for up to 20 min, when the reaction was stopped with 50 µl/well 0.5M $H_2SO_4$. A PHERAstar FS microplate reader (BMG Labtech) was used to measure absorption at 450 nm. Absorbance values from serially diluted samples were plotted and values which fell into the linear range of the curve were selected to calculate endpoint titres.

## RNA sequencing

Samples for RNA sequencing were obtained from 16 mice in one FACS sort. cDNA was prepared from sorted cells using the SMART-Seq v4 Ultra Low Input RNA Kit for Sequencing (Clontech # 634890) on the day of the sort following the manufacturer's protocol. 400 pg of cDNA per sample were used as input for the preparation of sequencing libraries with the Illumina Nextera XT kit (#FC-131–1096) following the manufacturer's instructions. The quality of the cDNA and libraries was assessed using Agilent Bioanalyser High Sensitivity DNA Chips (#5067–4626), and Qubit dsDNA High Sensitivity Assay Kit (Invitrogen #Q32854) on a Qubit 4 Fluorometer (Invitrogen). Six samples from young mice and six samples from aged mice passed all quality controls, were pooled onto two HiSeq sequencing lanes and sequenced as 100 bp single-end reads. RNA sequencing analysis was performed using the SeqMonk software package (Babraham Institute, https://www.bioinformatics.babraham.ac.uk/projects/seqmonk/) after trimming (Trim Galore v0.4.2) and alignment of reads to the reference mouse genome GRCm38 using HISAT2 (*Kim et al., 2015*). Reads were quantitated over exons and library size was standardized to 1 million reads, and then read counts were log2 transformed. Differentially expressed genes were determined by DESeq2 analysis using raw counts

(adjusted *p*-value cut-off p≤0.05) (*Love et al., 2014*). Principal component analysis was performed using 1000 genes with the largest variances, after normalisation for batch effects with RUVSeq (*Risso et al., 2014*).

To test for the differential expression of functionally related gene sets, a publicly available list of gene sets (Mouse_GO_AllPathways_with_GO_iea_December_24_2014_symbol.gmt.txt of Bader Lab EM_Genesets *Merico et al., 2010*) was filtered for categories containing less than 20 or more than 500 genes. Resulting gene sets were tested for differential expression between young and aged samples using Seqmonk Subgroup Statistics (Kolmogorov-Smirnov test, p<0.05, average absolute z-score >1, multiple testing correction). Genes in the Responsiveness to IFN-I pathway: *Stat1, Aim2, Pyhin1, Ifi204, Ifi203, Ifi202b, Ifi205, Gbp3, Gbp2, Ifnb1, Gbp6, Htra2, Ndufa13, Trex1, Pnpt1, Tgtp1, Irf1, Igtp, Ddx41, Tmem173, Gm4951, Iigp1, Ifit3, Ifit1*.

## Publicly available datasets

PBMC RNA-Seq data from GSE45735 were used to assess an individual's IFN response over time (*Henn et al., 2013*). Corresponding fastq files were obtained from SRA using the sratoolkit (https://www.ncbi.nlm.nih.gov/sra/) and aligned to GRCh38 using HISAT2 (*Kim et al., 2015*). Counting, at gene level, was performed with Rsubread (*Liao et al., 2019*). Variance stabilised normalisation (VSN, as implemented in DESeq2 *Huber et al., 2002*; *Love et al., 2014*) was applied to the counts to give an expression value per gene. To analyse the IFN-I response in these datasets we used the 'HALL-MARKS' IFN alpha response, which is a gene set curated by MSigDB at the Broad Institute based on experimentally derived expression data (*Subramanian et al., 2005*; *Liberzon et al., 2015*), and which we have used previously to identify and validate a role for IFN-I in the formation of ectopic lymphoid structures (*Denton et al., 2019*).

Microarray data from GSE74813 (*Franco et al., 2013*; *Nakaya et al., 2015*) were re-analysed as follows: Raw. CEL files were downloaded from GEO with the corresponding annotation data. CEL files were read into R via readAffy (*Gautier et al., 2004*) and were normalised using VSN (*Huber et al., 2002*). The dataset was then subsetted to select only individuals with paired day 0 and day 1 samples. Genes were selected as described in the results section. Antibody titres were made available upon request (*Nakaya et al., 2015*).

## Statistics

All mouse experiments were performed twice or more with 3–10 mice per group. Differences between experimental groups were assessed using the non-parametric Mann–Whitney or Kruskal-Wallis test combined with Dunn's multiple testing correction within the Prism v6 and v7 software (GraphPad). Outliers as determined by Tukey's outlier test within the Prism software were excluded from the analysis. Test Rank correlations were determined using Spearman's correlation coefficients (rho). p-values≤0.05 were considered statistically significant. For RNA sequencing analysis, sequencing reads were quantitated over exons and library size was standardized to 1 million reads, and then read counts were log2 transformed. Differentially expressed genes were determined by DESeq2 analysis using raw counts (adjusted *p*-value cut-off p≤0.05).

## Acknowledgements

We are grateful to Dr Wim Pierson for preparing Eα-GFP and his technical support, Dr Alice Denton for the mouse artwork in *Figure 7*, Dr Helder Nakaya for the phenotypic data that accompanied the human microarray dataset, Dr Anne O'Garra for the TCR7 TCR-Tg mice, Dr Oliver Bannard for provision of the XL-1 blue *E. coli* carrying a pTRCHis-Eα-GFP vector and Drs Geoff Butcher and Martin Turner for critical reading of the manuscript. We acknowledge the contribution of the Babraham Institute Biological Support Unit staff, who performed in vivo treatments of our animals and took care of animal husbandry. We thank the staff of the Babraham Flow Cytometry Facility for cell sorting and the Babraham Sequencing Facility for performing RNA sequencing. The authors are grateful to Dr Anne Corcoran for her input on the experiments with ageing animals. This study was supported by funding from the Biotechnology and Biological Sciences Research Council (BBS/E/B/000C0407, BBS/E/B/000C0427 and the Campus Capability Core Grant to the Babraham Institute), the European Research Council (637801 TWILIGHT), and the European Union's Horizon 2020 research and innovation programme 'ENLIGHT-TEN' under the Marie Skłodowska-Curie grant agreement No.:

675395. We gratefully acknowledge the participation of all NIHR BioResource Centre Cambridge volunteers, and thank the NIHR BioResource Centre Cambridge and staff for their contribution. We thank the National Institute for Health Research and NHS Blood and Transplant. The views expressed are those of the authors and not necessarily those of the NHS, the NIHR or the Department of Health & Social Care.

## Additional information

### Funding

| Funder | Grant reference number | Author |
|---|---|---|
| Biotechnology and Biological Sciences Research Council | BBS/E/B/000C0407 | Michelle A Linterman |
| Biotechnology and Biological Sciences Research Council | BBS/E/B/000C0408 | Michelle A Linterman |
| H2020 European Research Council | 637801 | Michelle A Linterman |
| H2020 Marie Skłodowska-Curie Actions | 675395 | Michelle A Linterman |

The funders had no role in study design, data collection and interpretation, or the decision to submit the work for publication.

### Author contributions

Marisa Stebegg, Data curation, Formal analysis, Investigation, Visualization, Methodology; Alexandre Bignon, Data curation, Formal analysis, Investigation, Methodology; Danika Lea Hill, Alyssa Silva-Cayetano, Edward Carr, Investigation, Visualization, Methodology; Christel Krueger, Louis Boon, Martin S Zand, James Dooley, Jonathan Clark, Investigation, Methodology; Ine Vanderleyden, Silvia Innocentin, Methodology; Jiong Wang, Investigation; Adrian Liston, Supervision, Methodology; Michelle A Linterman, Conceptualization, Data curation, Formal analysis, Supervision, Funding acquisition, Visualization, Methodology, Project administration

### Author ORCIDs

Marisa Stebegg https://orcid.org/0000-0001-8434-8271
Alyssa Silva-Cayetano http://orcid.org/0000-0003-0533-1629
Christel Krueger http://orcid.org/0000-0001-5601-598X
Edward Carr http://orcid.org/0000-0001-9343-4593
Michelle A Linterman https://orcid.org/0000-0001-6047-1996

### Ethics

Human subjects: All human blood and tissue was collected in accordance with the latest revision of the Declaration of Helsinki and the Guidelines for Good Clinical Practice (ICH-GCP). The seasonal UK influenza vaccination cohort was collected with UK local research ethics committee approval (REC reference 14/SC/1077), using the facilities of the Cambridge Bioresource (REC reference 04/Q0108/44). Written informed consent was received from all volunteers.
Animal experimentation: All mouse experimentation was approved by the Babraham Institute Animal Welfare and Ethical Review Body. Animal husbandry and experimentation complied with existing European Union and United Kingdom Home Office legislation and local standards (PPL: P4D4AF812).

### Decision letter and Author response

Decision letter https://doi.org/10.7554/eLife.52473.sa1
Author response https://doi.org/10.7554/eLife.52473.sa2

## Additional files

### Supplementary files
• Supplementary file 1. Key Resources Table.

• Transparent reporting form

### Data availability

Source data files are included with this manuscript. The RNA sequencing data generated for this study have been deposited at GEO (GSE133148). Further data in support of our findings are available from the corresponding author upon request, excepting the antibody data for GSE74813 which should be requested from its authors (Nakaya et al., 2015).

The following dataset was generated:

| Author(s) | Year | Dataset title | Dataset URL | Database and Identifier |
|---|---|---|---|---|
| Hill D, Bignon A, Linterman M | 2020 | Transcriptiomics of cDC2 from young adult and aged mice after immunisation | https://www.ncbi.nlm.nih.gov/geo/query/acc.cgi?acc=GSE133148 | NCBI Gene Expression Omnibus – GEO, GSE133148 |

The following previously published datasets were used:

| Author(s) | Year | Dataset title | Dataset URL | Database and Identifier |
|---|---|---|---|---|
| Henn AD, Wu S, Qiu X, Ruda M, Stover M, Yang H, Liu Z, Welle SL, Holden-Wiltse J, Wu H, Zand MS | 2013 | Changes in PBMC gene expression profiles after influenza vaccination in healthy human subjects | https://www.ncbi.nlm.nih.gov/Traces/study/?acc=SRP020492 | SRP020492, SRP020492 |
| Nakaya HI, Pulendran B | 2015 | Time Course of Adults Vaccinated with Influenza TIV Vaccine during 2010/11 Flu Season (HIPC cohort) | https://www.ncbi.nlm.nih.gov/geo/query/acc.cgi?acc=GSE74813 | NCBI Gene Expression Omnibus, GSE74813 |

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
