## [Decision Letter]

Thank you for submitting your article "Restoration of type I interferon signalling in dendritic cells rescues T follicular helper cell formation in ageing" for consideration by *eLife*. Your article has been reviewed by Satyajit Rath as the Senior Editor, a Reviewing Editor, and two reviewers. The reviewers have opted to remain anonymous.

The reviewers have discussed the reviews with one another and the Reviewing Editor has drafted this decision to help you prepare a revised submission.

Summary:

A number of studies have demonstrated that vaccine responses are impaired in older patient populations leaving them vulnerable to infections such as influenza. As a result, it is a topic of current interest to discover the details of how this occurs in order to find strategies to improve vaccine response in aged cohorts. In this study, Stebegg et al., demonstrate that Tfh responses are reduced in both aged humans and mice and suggest that impaired type-I interferon signaling in cDC2 results in reduced CD80 and CD86, which in turn affects Tfh formation and antibody responses. Overall, the paper is well written and covers a topic of significant interest. However, there are a few issues that need to be cleaned up.

Essential revisions:

1) Impaired DC activation/Tfh generation in ageing, and the importance of type-I IFN signaling in DCs for Tfh generation, have been previously published. In this regard, the current study mechanistically links previous observation to explain the defective Tfh generation in ageing by finding the reduced type-I IFN signaling in DCs in aged mice. Given that DCs from aged mice were able to respond normally to exogenous IFNa, the reduced amount of IFNa/b produced in aged mice after immunization or infection could explain the defective Tfh generation. What type of cells produce IFNa/b in response to immunization for proper cDC2 activation? The IFNb producing cells should be identified in Figure 5F.

2) Figure 8: The authors use Imiquimod to induce IFN signaling and restore cDC2 function in aged mice. While these data are of interest Imiquimod is not a specific inducer of interferon signaling and also acts via a range of other pathways to induce other Tfh relevant factors such as IL-6. (Br J Dermatol. 2007 Dec;157 Suppl 2:8-13. Imiquimod: mode of action. Schön MP1, Schön M.). This is brought into particular focus by the title of the paper. The CD11c conditional IFNαR mouse used in Figure 7 should be used to confirm if the effect of imiquimod is dependent on IFN signaling in this context.

1) In Figure 3G, # of GFP+cDC2 in draining LNs was reduced after immunization of aged mice with Ea-GFP. Does this mean that ability of cDC2s in aged mice to endocytose antigens is defective or that total number of cDC2 in draining LNs of aged mice is decreased?

2) Antigen-specific antibody response (High affinity vs low affinity in case of NP immunization) should be shown in Figure 2, Figure 3, Figure 7, Figure 8.

3) Does Imiquimod also have a Tfh enhancing effect in young mice? Some caution has to be used when an agent that enhances responses in all populations is interpreted as specifically restoring a sub-population specific defect.

4) In Figure 3 the authors note that the aged cDC2 have reduced MHCII, CD80, CD86 and CD40. In Figure 4 they attempt to unpick this by using CTLA-4Ig in Figure 4 which reduces T-cell proliferation. Does it also reduce the% Tfh? However, while it is possible to use CTLA-4Ig to demonstrate the broad point that blocking CD80 and CD86 impacts T-cell proliferation and Tfh formation it not clear how this provides any evidence that loss of CD80 and CD86 on cDC2 is the specific cause of low of Tfh in aged mice. These data could be moved to the supplemental material. It is reasonable to speculate that the changes to CD86 and CD80 seen in Figure 3 may have a role, and it has previously been demonstrated that even modest changes to CD28 signaling may be enough to alter Tfh formation (Wang, 2015), but the authors should be aware that they have not proved this experimentally.

5) Figure 7: E and G. Representative data are shown from only one group and so does not allow any comparison. Please include examples from both groups.

---

## [Author Response]

Summary:A number of studies have demonstrated that vaccine responses are impaired in older patient populations leaving them vulnerable to infections such as influenza. […]. Overall, the paper is well written and covers a topic of significant interest. However, there are a few issues that need to be cleaned up.

We appreciate this supportive review and have revised the manuscript according to the suggestions. The details of how the comments have been addressed are included below under each point. Alterations are underlined in the text of the manuscript, and all relevant text and figures are included in the response to reviewers under each point.

Essential revisions:1) Impaired DC activation/Tfh generation in ageing, and the importance of type-I IFN signaling in DCs for Tfh generation, have been previously published. In this regard, the current study mechanistically links previous observation to explain the defective Tfh generation in ageing by finding the reduced type-I IFN signaling in DCs in aged mice. Given that DCs from aged mice were able to respond normally to exogenous IFNa, the reduced amount of IFNa/b produced in aged mice after immunization or infection could explain the defective Tfh generation. What type of cells produce IFNa/b in response to immunization for proper cDC2 activation? The IFNb producing cells should be identified in Figure 5F.

To identify the cellular source of *Ifnb1* production after immunisation we sorted T cells, B cells, macrophages, plasmacytoid DCs (pDC), cDC1 and cDC2/Langerhans cells from the lymph node of either unimmunised or immunised mice. RT-qPCR shows that pDCs and CD64^+^F4/80^+^ macrophages produced the highest levels of *Ifnb1* after immunisation. This potentially implicates changes in the number or function of these cells in ageing that leads to a reduced IFN-I response in cDC2s. Interestingly, there is an age-associated decrease in pDC numbers which may contribute to the phenotype observed in aged mice. In bone marrow (BM) chimeras where aged mice were irradiated and reconstituted with BM from younger adult mice, the reduction in pDC numbers persists in the aged host, indicating a cell-extrinsic cause of reduced pDCs in aged mice. In these chimeras, the number of GFP^+^ cDC2 is diminished, as was their expression of CD80/86, suggesting that the aged microenvironment, and potentially the reduction of IFN-I-producing pDC contributes to impaired dendritic cell activation in ageing. All these data have been included in Figure 4. We have also included a section in the Material and Methods section to describe how the cell sort was performed.

To further exclude any contribution of cell-intrinsic factors to the reduced IFN-I response of cDC2s in aged mice, we compared the expression of Interferon stimulated genes (ISGs) in cDC2s from adult and aged mice upon ex vivo treatment with low levels of IFNα. These data are included in Figure 4 and revealed that the upregulation of the ISGs *Ifit1* and *Mx1* in cDC2s are intact in aged mice, consistent with the pSTAT1 data provided in the original version of the paper. This supports the interpretation of the data that aged cDC2s are capable of responding to IFN-I and do not exhibit cell-intrinsic defects in IFN signalling.

These new results have been incorporated into the manuscript as follows:

Subsection “The cDC2 response to IFN-I is reduced in aged mice”:

“STAT1 phosphorylation levels and the upregulation of the ISGs *Ifit1* and *Mx1* in cDC2s were intact in aged mice upon ex vivo treatment with low levels of IFNα (Figure 4G-J). […] Together, this suggests that the poor activation of cDC2s in the LNs of aged animals could be driven by reduced numbers of IFN-I-producing pDCs due to age-related changes in their microenvironment.”

Discussion section:

“Ageing has been reported to impair DC activation (Agrawal et al., 2007; Moretto, Lawlor and Khan, 2008), the production of IFN-I cytokines by pDCs in both humans and mice (Stout-Delgado et al., 2008b; Panda et al., 2010; Sridharan et al., 2011; Agrawal, 2013; Agrawal, Agrawal and Gupta, 2017), and the formation of Tfh cells (Garcia and Miller, 2001b; Eaton et al., 2004; Lefebvre et al., 2012; Linterman, 2014; Gustafson, Weyand and Goronzy, 2018; Nikolich-Žugich, 2018). In addition, Brahmakshatriya and colleagues have demonstrated that transferring activated, in vitro BM-derived DCs into aged mice can boost both the GC and Tfh cell response upon immunisation (Brahmakshatriya et al., 2017). Our study demonstrates that age-associated defects in the early induction of IFN-I expression, probably by pDCs and macrophages, results in impaired expression of co-stimulatory molecules on cDC2s.”

Figure 4 legend:

“(I-J) *Ifit1* (I) and *Mx1* (J) mRNA levels were determined in sorted GFP^+^ CD11b^+^ cDC2s by RT-qPCR upon ex vivo treatment of LN cells with 50 U murine IFNα for 3 hours. […] In (N) bar heights correspond to the mean and error bars represent standard deviation. *P*-values were determined using the Mann-Whitney test.”

Subsection “Bone marrow chimeras”:

**“**Recipient mice (2-month-old CD45.1^+^ C57BL/6 SJL or 2-month-old and 21-month-old C57BL/6 mice) were irradiated with 800-1000 rad in two doses and reconstituted viaintravenous injection with 2-4 × 10^6^ BM cells isolated from donor mice (2-3-month-old CD45.1^+^ C57BL/6 SJL, 3-month-old or 23-month-old C57BL/6 mice). BM chimeras were administered neomycin in their drinking water for the first four weeks after BM transfers and were used for experiments eight weeks after successful reconstitution.”

Subsection “Fluorescence-activated cell sorting (FACS)”:

“To determine the IFN-I responsiveness of DCsex vivo, ~2 x 10^6^ cells isolated from the inguinal LNs of naïve mice were seeded into sterile round-bottom 96-well plates in 200 µl complete RPMI. […] RNA for RT-qPCR was isolated from cell pellets using Qiagen’s RNeasy Mini or Micro Kit (#74104 and #74004).”

2) Figure 8: The authors use Imiquimod to induce IFN signaling and restore cDC2 function in aged mice. While these data are of interest Imiquimod is not a specific inducer of interferon signaling and also acts via a range of other pathways to induce other Tfh relevant factors such as IL-6. (Br J Dermatol. 2007 Dec;157 Suppl 2:8-13. Imiquimod: mode of action. Schön MP1, Schön M.). This is brought into particular focus by the title of the paper. The CD11c conditional IFNαR mouse used in Figure 7 should be used to confirm if the effect of imiquimod is dependent on IFN signaling in this context.

We agree with the reviewers that imiquimod may also act via IFN-I-independent mechanisms to promote cDC2 and Tfh cells in aged mice and that this is important for the interpretation of the data presented in this paper. We have done two sets of experiments to address this key point.

1) *Ifnar1*^-/-^ miceand controls were treated with imiquimod to determine whether effects of imiquimod on DC numbers and activation early after immunisation are also mediated by IFN-I signalling. These data show that the effect of imiquimod on cDC2 numbers and CD80/86 expression are largely IFN-I driven. These data are now included in Figure 7H-J, and discussed in subsection “The TLR7 agonist imiquimod boosts cDC2s and Tfh cell numbers”.

“Topical imiquimod treatment of *Ifnar1*^-/-^ and *Ifnar1*^+/+^ mice revealed that the enhancing effects of imiquimod on cDC2 numbers and activation were largely, but not completely, dependent on IFN-I signalling (Figure7H-J). This demonstrates that imiquimod treatment can boost the reduced IFN-I response, and revert the numerical and co-stimulatory defects observed in cDC2s from aged mice.”

Figure 7 legend:

“(H-J) 2 month old *Ifnar1*^-/-^ and *Ifnar1*^+/+^ mice were immunised subcutaneously with Eα-GFP in IFA and some of the mice were additionally treated with imiquimod cream over their immunisation sites. (H) 22 hours later the number of GFP^+^ CD11b^+^ cDC2 cells in the draining lymph nodes (LNs) were quantified. (I-J) Quantitation of median fluorescence intensity (MFI) levels of CD86 (I) and CD80 (J) on the surface of these GFP^+^ CD11b^+^ cDC2s. Bar graphs show the results of one of two independent experiments (B-G; n=6 per group/experiment) or the pooled results from two experiments (H-J; n=3-11 per group). Bar height corresponds to the median, and each circle represents one biological replicate. In (B-G) *p*-values were determined using the Mann-Whitney test. In (H-J) *p*-values were determined by comparing each group to the “*Ifnar1*^+/+^ no imiquimod control” group using the Kruskal Wallis test with Dunn’s multiple testing correction. Supporting data are shown in Figure 7—figure supplement 1.

2) To investigate whether the effect of imiquimod on Tfh cell formation is IFN-I-independent, we conducted the experiment suggested by the reviewer and treated 2 month old *Ifnar1^fl/fl^Itgax^cre/+^*or *Ifnar1^fl/fl^Itgax^+/+^* control mice with imiquimod upon immunisation and analysed Tfh cell formation seven days later. These data show that imiquimod is able to boost Tfh cells to the same extent in *Ifnar1^fl/fl^Itgax^cre/+^*mice as in control animals. These data demonstrate that there are IFN-independent effects of imiquimod that are able to boost Tfh cells in younger mice, these data are now included as Figure 8G-H and discussed on page 14. Because of these results we have revised the manuscript to ensure that all conclusions and interpretations are consistent with these new data, this includes revising the Title, Abstract and Discussion section of the paper.

Title: Rejuvenating conventional dendritic cells and T follicular helper cell formation after vaccination

Abstract: […] Here, we demonstrate that older people and aged mice have impaired Tfh cell differentiation upon vaccination. This deficit is preceded by poor activation of conventional dendritic cells type 2 (cDC2) due to reduced type 1 interferon signalling. Importantly, the Tfh and cDC2 cell response can be boosted in aged mice by treatment with a TLR7 agonist. This demonstrates that age-associated defects in the cDC2 and Tfh cell response are not irreversible and can be enhanced to improve vaccine responses in older individuals.

Subsection “The TLR7 agonist imiquimod boosts cDC2s and Tfh cell numbers” “Topical imiquimod treatment potently enhanced total and 1W1K-specific Tfh cell numbers in both 2-3- and 22-24-month-old mice seven days after immunisation (Figure 8A-F). […] Imiquimod treatment did not affect serum levels of antigen-specific antibodies at this early time-point when most antibodies come from the extrafollicular plasmablast response (Figure 8—figure supplement 1A-C) (MacLennan et al., 2003). “

Figure 8 legend:

“Imiquimod rejuvenates Tfh cell differentiation. 2-3-month-old and 22-24-month-old C57BL/6 mice (A-F, I-K) or 2 month old *Ifnar1^fl/fl^Itgax^cre/+^*and*Ifnar1^fl/fl^Itgax^+/+^*littermate controls (G-H) were all subcutaneously immunised with NP-1W1K in Alum and then either topically treated with imiquimod cream over their immunisation sites (“Imiquimod”, *Ifnar1^fl/fl^Itgax^cre/+^*and*Ifnar1^fl/fl^Itgax^+/+^*mice)or left untreated (“Untreated”). […] *P*-values were determined using the Mann-Whitney test. Supporting data are shown in Figure 8—figure supplement 1.”

Discussion section:

“In an attempt to enhance cDC2 and Tfh cell responses upon vaccination, we applied a cream containing the TLR7-agonist imiquimod, which has been shown to induce IFN-Is (Chen et al., 1988; Bottrel et al., 1999; Sauder, 2003) to the skin of mice upon immunisation. […] Together, this indicates that the TLR7 agonist imiquimod can boost cDC2 and Tfh cells using more than one mechanism, reinforcing its potential as a vaccine adjuvant.”

1) In Figure 3G, # of GFP+cDC2 in draining LNs was reduced after immunization of aged mice with Ea-GFP. Does this mean that ability of cDC2s in aged mice to endocytose antigens is defective or that total number of cDC2 in draining LNs of aged mice is decreased?

In our experiments we see a reduced number of total cDC2 draining to the LN, which was enhanced by application of imiquimod. To clarify this point, we have now included graphs showing the number of total, as well as GFP^+^ CD11b^+^ cDC2s in Figures 3G and 7D, which are mentioned Subsection “T cell priming is impaired in aged mice” and Subsection “The TLR7 agonist imiquimod boosts cDC2s and Tfh cell numbers”.

Subsection “T cell priming is impaired in aged mice”:

“One day after immunisation, aged mice had half the number of total and GFP^+^ CD11b^+^ cDC2s compared to younger controls (Figure 3G-H).”

Figure 3 legend:

“(G-P) 2-3-month-old mice and 22-24-month-old were immunised subcutaneously with Eα-GFP in IFA. Antigen-bearing GFP^+^ and antigen-presenting Y-Ae^+^ dendritic cells (DCs) in draining LNs were analysed 22 hours after immunisation (n=7-10 per group/experiment). (G-H) Quantitation of total (G) and GFP+ (H) CD11b^+^ type 2 conventional DCs (cDC2s). Bar graphs show the results of one of at least two independent experiments. Bar height corresponds to the median, and each circle represents one biological replicate. *P*-values were determined using the Mann-Whitney test.”

Subsection “The TLR7 agonist imiquimod boosts cDC2s and Tfh cell numbers”:

“This was associated with a two- to three-fold increase in the number of total and GFP^+^ CD11b^+^ cDC2s in draining LNs (Figure 7D-E).”

Figure 7 legend:

“(D-E) Flow cytometric quantitation of total (D) and GFP^+^ (E) CD11b^+^ cDC2 cells in the draining lymph nodes (LNs) of 22-24-month-old mice with or without imiquimod treatment. Bar graphs show the results of one of two independent experiments (n=6 per group/experiment). Bar height corresponds to the median, and each circle represents one biological replicate. *P*-values were determined using the Mann-Whitney test.”

2) Antigen-specific antibody response (High affinity vs low affinity in case of NP immunization) should be shown in Figure 2, Figure 3, Figure 7, Figure 8.

These data have now been included in Figure 2E-G, Figure 3—figure supplement 1A-C, Figure 6—figure supplement 1A-C and Figure 8—figure supplement 1A-C. It is important to note, however, that the early time points of our experiments (7-10 days after immunisation) will be largely measuring antibody produced from the extrafollicular plasmablast response rather than from the germinal centre reaction.

Subsection “Tfh cell and GC responses are impaired in ageing”:

“However the number of GC B cells was ten-fold lower in the aged mice ten days after immunisation, compared to younger adult mice (Figure 2C). This corresponded to a reduction in GC size (Figure 2D) and reduced levels of antigen-specific antibodies in the serum of aged mice (Figure 2E-G).”

Figure 2 legend:

“(E-G) Levels of NP-specific IgG1 antibodies in the serum of 2-3-month-old and 22-24-month-old mice 10 days after immunisation with NP-1W1K in Alum as determined by ELISA. (E) Serum levels of NP20-specific IgG1 antibodies. (F) Serum levels of high-affinity NP7-specific IgG1 antibodies. (G) Ratio of NP20/NP7-specific IgG1 antibodies in the serum as a measure of antibody affinity maturation. Bar graphs show the results of one of two independent experiments (n=8 per group/experiment). Bar height corresponds to the median, and each circle represents one biological replicate. *P*-values were determined using Mann-Whitney testing.”

Subsection “Tfh cell and GC responses are impaired in ageing”::

“In addition to defects in T cell priming, Tfh cell differentiation of OTII^+^ CD4^+^ T cells isolated from 2-3-month-old mice was reduced three-fold ten days after immunisation in aged mice compared to younger recipients (Figure 3E-F), which was associated with reduced levels of antigen-specific antibodies in the serum of these mice (Figure 3—figure supplement 1A-C).”

Figure 3—figure supplement 1 legend:

“(A-C) 5 × 10^4^ OVA-specific (CD45.1^+^TCRVα2^+^CD4^+^) OTII cells were adoptively transferred into 2-3-month-old and 22-24-month-old C57BL/6 recipients, which were subsequently immunised subcutaneously with NP-OVA in Alum in the hind flank. Ten days later, NP-specific IgG1 antibody response were quantified by ELISA in the serum of these mice (n=4-6 per group/experiment). (A) Serum levels of NP20-specific IgG1 antibodies. (B) Serum levels of high-affinity NP7-specific IgG1 antibodies. (C) Ratio of NP20/NP7-specific IgG1 antibodies in the serum as a measure of antibody affinity maturation. Graphs show the results of one of two independent experiments. Bar height corresponds to the median, and each circle represents one biological replicate. *P*-values were determined using the Mann-Whitney test.”

Subsection “Lack of IFN-I signalling in DCs results in impaired Tfh cell formation”:

“In the absence of IFNAR1 on DCs, there was a defect in Tfh cell formation seven days after immunisation (Figure 6E-H), without affecting serum levels of antigen-specific antibody at this early time point (Figure 6—figure supplement 1A-C).“

Figure 6—figure supplement 1 legend:

“Figure 6—figure supplement 1: Lack of IFN-I signalling in DCs does not affect early antigen-specific antibody responses. (A-C) Quantitation of NP-specific IgG1 antibody responses in the serum of *Ifnar1^fl/fl^Itgax^cre/+^* or *Ifnar1^fl/fl^Itgax^+/+^*control mice seven days after immunisation with NP-1W1K in Alum by ELISA (n=4-9 per group/experiment). (A) Serum levels of NP20-specific IgG1 antibodies. (B) Serum levels of high-affinity NP7-specific IgG1 antibodies. (C) Ratio of NP20/NP7-specific IgG1 antibodies in the serum as a measure of antibody affinity maturation. Bar graphs show the results of one of two experimental repeats. Bar height corresponds to the median, and each circle represents one biological replicate. *P*-values were determined using the Mann-Whitney test.”

Subsection “The TLR7 agonist imiquimod boosts cDC2s and Tfh cell numbers”:

“This was linked with a small, but significant, increase in the number of GC B cells in aged mice, but not in younger animals (Figure 8G-H), and did not affect serum levels of antigen-specific antibodies at this early time-point when most antibodies come from the extrafollicular plasmablast response (Figure 8—figure supplement 1A-C) (MacLennan et al., 2003).”

Figure 8—figure supplement 1 legend:

“Figure 8—figure supplement 1: Imiquimod treatment does not affect early antigen-specific antibody responses. (A-C) 22-24-month-old mice were immunised subcutaneously with NP-1W1K in Alum. Half of the mice were topically treated with imiquimod cream over their immunisation sites. Seven days later, serum levels of antigen-specific IgG1 antibodies were determined by ELISA. (A) Serum levels of NP20-specific antibodies. (B) Serum levels of high-affinity NP7-specific antibodies. (C) Ratio of NP20/NP7-specific antibodies in the serum as a measure of antibody affinity maturation. Bar graphs show the pooled results of two experiments (n=6 per group/experiment). Bar height corresponds to the median, and each circle represents one biological replicate. *P*-values were determined using the Mann-Whitney test.”

We have also included subsection ““Enzyme-linked immunosorbent assay (ELISA)” to describe how ELISAs were performed.

“ELISA plates (Thermo Fisher Scientific 96F Maxisorp #456537) were coated overnight at 4°C with 10 µg/ml NP20-BSA (Biosearch Technologies #N-5050H-100) or 2.5 µg/ml NP7-BSA (Biosearch Technologies #N-5050L-100) in PBS. […] Absorbance values from serially diluted samples were plotted and values which fell into the linear range of the curve were selected to calculate endpoint titres.”

3) Does Imiquimod also have a Tfh enhancing effect in young mice? Some caution has to be used when an agent that enhances responses in all populations is interpreted as specifically restoring a sub-population specific defect.

We have included data from young mice treated with imiquimod at the time of immunisation in Figure 7—figure supplement 1A-F and Figure 8C, F, K and have modified subsection “The TLR7 agonist imiquimod boosts cDC2s and Tfh cell numbers” to reflect these changes. These data show that imiquimod enhances Tfh cell differentiation irrespective of age, but only enhanced GC B cell numbers in aged mice.

“Imiquimod treatment increased expression of the ISGs *Ifit1* and *Mx1* in antigen-bearing cDC2s from both 2-3- and 22-24-month-old mice compared to age-matched no-imiquimod controls one day after immunisation (Figure 7B-C; Figure 7—figure supplement 1A-B). This was associated with a two- to three-fold increase in the number of total and GFP^+^ CD11b^+^ cDC2s in draining LNs (Figure 7D-E; Figure 7—figure supplement 1C-D) and with a two-fold increase in CD80 and CD86 on their surfaces (Figure 7F-G, Figure 7—figure supplement 1E-F). […] In addition, imiquimod potently enhanced total and 1W1K-specific Tfh cell numbers in both 2-3- and 22-24-month-old mice seven days after immunisation (Figure 8A-F).

Figure 7—figure supplement 1 legend:

“Figure 7—figure supplement 1: Imiquimod induces IFN-I signalling and boosts cDC2 responses. 2-3-month-old mice were immunised subcutaneously with either Eα-GFP in IFA. […] Bar height corresponds to the median, and each circle represents one biological replicate. *P*-values were determined using the Mann-Whitney test.”

Figure 8 legend:

“Figure 8: Imiquimod rejuvenates Tfh cell differentiation. 2-3-month-old and 22-24-month-old C57BL/6 mice (A-F, I-K) or *Ifnar1^fl/fl^Itgax^cre/+^*and*Ifnar1^fl/fl^Itgax^+/+^*littermate controls (G-H) were all subcutaneously immunised with NP-1W1K in Alum and then either topically treated with imiquimod cream over their immunisation sites (“Imiquimod”, *Ifnar1^fl/fl^Itgax^cre/+^*and*Ifnar1^fl/fl^Itgax^+/+^*mice)or left untreated (“Untreated”). […] Supporting data are shown in Figure 8—figure supplement 1.”

4) In Figure 3 the authors note that the aged cDC2 have reduced MHCII, CD80, CD86 and CD40. In Figure 4 they attempt to unpick this by using CTLA-4Ig in Figure 4 which reduces T-cell proliferation. Does it also reduce the% Tfh? However, while it is possible to use CTLA-4Ig to demonstrate the broad point that blocking CD80 and CD86 impacts T-cell proliferation and Tfh formation it not clear how this provides any evidence that loss of CD80 and CD86 on cDC2 is the specific cause of low of Tfh in aged mice. These data could be moved to the supplemental material. It is reasonable to speculate that the changes to CD86 and CD80 seen in Figure 3 may have a role, and it has previously been demonstrated that even modest changes to CD28 signaling may be enough to alter Tfh formation (Wang, 2015), but the authors should be aware that they have not proved this experimentally.

As suggested, we have moved Figure 4 to Supplementary Figure 3—figure supplement 1H-L and changed subsection “The cDC2 response to IFN-I is reduced in aged mice” to clarify our interpretation of the data, particularly with respect to Wang, 2015.

“By partially blocking these co-stimulatory ligands using a CTLA4-Ig fusion protein in vivo, we observed a dose-dependent decrease in early T cell proliferation (Figure 3—figure supplement 1K-L). Moreover, Wang et al. had previously demonstrated that reducing the magnitude of CD28 signalling in vivo impairs Tfh cell differentiation (C. J. Wang et al., 2015). This implicates the age-associated reduction in the expression of CD80/CD86 on cDC2s as a likely factor that contributes to impaired T cell priming and Tfh cell formation in aged mice.”

5) Figure 7: E and G. Representative data are shown from only one group and so does not allow any comparison. Please include examples from both groups.

We have now included examples from both groups in this figure (now Figure 6E and G).

Figure 6 legend:

“(E-H) Flow cytometric analysis (E, G) and quantitation (F, H) of total (E-F) and 1W1K-I-Ab^+^ (G-H) CXCR5^hi^PD-1^hi^Foxp3^-^CD4^+^ T follicular helper (Tfh) cells isolated from *Ifnar1^fl/fl^Itgax^cre/+^* or *Ifnar1^fl/fl^Itgax^+/+^*control mice seven days after immunisation with NP-1W1K in Alum (n=8-9 per group/experiment). Bar graphs show the results of one of two experimental repeats. Bar height corresponds to the median, and each circle represents one biological replicate. *P*-values were determined using the Mann-Whitney test.”